# Acentrosomal spindles assemble from branching microtubule nucleation near chromosomes in *Xenopus laevis* egg extract

Bernardo Gouveia[1,6], Sagar U. Setru [2,6], Matthew R. King[3], Aaron Hamlin[3], Howard A. Stone [4], Joshua W. Shaevitz [2,5] & Sabine Petry [3] ✉

Microtubules are generated at centrosomes, chromosomes, and within spindles during cell division. Whereas microtubule nucleation at the centrosome is well characterized, much remains unknown about where, when, and how microtubules are nucleated at chromosomes. To address these questions, we reconstitute microtubule nucleation from purified chromosomes in meiotic *Xenopus* egg extract and find that chromosomes alone can form spindles. We visualize microtubule nucleation near chromosomes using total internal reflection fluorescence microscopy to find that this occurs through branching microtubule nucleation. By inhibiting molecular motors, we find that the organization of the resultant polar branched networks is consistent with a theoretical model where the effectors for branching nucleation are released by chromosomes, forming a concentration gradient that spatially biases branching microtbule nucleation. In the presence of motors, these branched networks are ultimately organized into functional spindles, where the number of emergent spindle poles scales with the number of chromosomes and total chromatin area.

Microtubules originate from centrosomes, chromosomes, and spindle microtubules in dividing eukaryotic cells to form mitotic and meiotic spindles. Chromosomal microtubule generation is particularly critical in cells that do not contain centrosomes, such as plant cells and meiotic egg cells in animals[1–4]. While microtubule nucleation from centrosomes has been well studied, it remains poorly understood how microtubules are generated around chromosomes because spindle microtubules can neither be resolved from one another nor can their exact origins be determined in cells. Thus, it remains a fundamental question in cell biology to understand where, when, and how microtubules are nucleated at chromosomes to build a spindle that successfully captures and segregates chromosomes during cell division.

Experiments in meiotic *Xenopus laevis* egg extracts have demonstrated that chromatin alone can generate spindles and tune a

spindle's size and shape[5–7]. Though revelatory, in lieu of actual chromosomes these studies used bacterial DNA on beads to form chromatin beads, which have different shapes and different amounts of chromatin compared to bona fide chromosomes. The studies also lacked kinetochores, the landing pads on chromosomes where microtubules that make up kinetochore fibers bind before pulling sister chromatids apart. Indeed, experiments in cultured mitotic cells have shown that microtubules also form in the vicinity of kinetochores during prometaphase[8–15], and in later stages at kinetochores on chromosomes unattached to existing spindle microtubules, as well as far away from the spindle equator, suggesting a back-up mechanism to capture unaligned chromosomes[11,12]. Microtubule nucleation near kinetochores has not been observed directly in a meiotic system, where it is perhaps even more important than in mitosis given the lack

[1]Department of Chemical and Biological Engineering, Princeton University, Princeton, NJ 08544, USA. [2]Lewis-Sigler Institute for Integrative Genomics, Princeton University, Princeton, NJ 08544, USA. [3]Department of Molecular Biology, Princeton University, Princeton, NJ 08544, USA. [4]Department of Mechanical and Aerospace Engineering, Princeton University, Princeton, NJ 08544, USA. [5]Department of Physics, Princeton University, Princeton, NJ 08544, USA. [6]These authors contributed equally: Bernardo Gouveia, Sagar U. Setru. ✉e-mail: spetry@princeton.edu

of centrosomes that have traditionally been considered the origin of kinetochore fibers.

One way to generate microtubules around chromosomes is via the RanGTP gradient[16]. The chromatin-bound guanine nucleotide exchange factor RCC1 equips Ran with GTP. RanGTP then releases spindle assembly factors (SAFs) sequestered by importin proteins, allowing SAFs to promote microtubule nucleation around chromosomes, as has been shown in *Xenopus* egg extracts using chromatin beads[17] or sperm chromatin[18–22], and in mitotic cells[23]. Cytoplasmic RanGAP1 deactivates RanGTP in a spatially uniform manner[24]. This combination of a localized source and homogeneous degradation creates RanGTP and SAF gradients centered at chromosomes.

A critical SAF is the protein TPX2[25], which along with the protein complex augmin, the γ-tubulin ring complex (γ-TuRC), and XMAP215, promotes the nucleation of new microtubules from the surface of existing microtubules[26]. This process, known as branching microtubule nucleation, generates the majority of spindle microtubules in mitotic cells[27] and *Xenopus* egg extracts[28]. Yet, where exactly branching microtubule nucleation occurs around chromosomes remains an open question.

There could be other microtubule nucleation pathways at play that originate from chromosomes. For example, the chromosomal passenger complex (CPC) at the centromeres of chromosomes[29] promotes microtubule growth by inhibiting the microtubule depolymerase MCAK, thereby promoting microtubule polymerization in *Xenopus* egg extracts[30]. It was shown that spindles can assemble around sperm nuclei in a manner independent of RanGTP but that requires the CPC[31]. The degree to which chromosomal microtubule nucleation differentially depends on RanGTP and the CPC, and whether branching microtubule nucleation occurs, remain to be determined.

Lastly, most quantitative studies to date have focused only on how microtubule nucleation sustains the steady-state metaphase spindle[28,32–34], but do not address the question of how a spindle builds its microtubule network starting from scratch at the end of interphase. Similarly, simulations have focused on either the steady-state metaphase spindle[32] or how centrosomes search for and capture kinetochores[35,36]. Currently, no quantitative model that starts with a realistic initial condition exists to describe how microtubules nucleate from chromosomes in the early stages of acentrosomal spindle assembly.

To explore these questions, we combine experiments that reconstitute microtubule nucleation from purified chromosomes in meiotic *Xenopus* egg extract with a mathematical model of branching nucleation in a SAF gradient. By directly visualizing microtubule nucleation at chromosomes using total internal reflection fluorescence microscopy (TIRFM), we reveal where individual microtubules nucleate, at what distances and orientations with respect to chromosomes they form, and how this is influenced by the amount of chromatin and the number of chromosomes. By comparing experimental results to our model, we find that SAF-mediated branching microtubule nucleation in the vicinity of chromosomes provides the main source of microtubules for acentrosomal spindles in *Xenopus* egg extract. We then establish how the structures of functional spindles that self-organize around purified chromosomes change with the amount of chromatin and the number of chromosomes.

## Results

### Microtubule-dependent microtubule nucleation is initiated near chromosomes

It is challenging to directly visualize the nucleation of individual microtubules near chromosomes in cells because the spindle becomes dense with microtubules within minutes[7,37,38]. To overcome this obstacle, we reconstituted chromosomal microtubule nucleation ex vivo. Briefly, we purified mitotic chromosomes with kinetochores labeled with GFP-CENPA from cultured mitotic HeLa cells. To enable live visualization of individual microtubules nucleating and growing at the onset of spindle assembly, we attached purified, DAPI-stained chromosomes to the functionalized and passivated glass coverslip of a microscope flow chamber ("Methods"). We then added *Xenopus* egg extract supplemented with fluorescent tubulin and EB1 to label microtubules and their growing plus-ends, respectively (Fig. 1a). The action of molecular motors was inhibited using the drug vanadate.

Using 4-color time-lapse TIRFM, we observed polar microtubule networks forming near chromosomes (Fig. 1b, Supplementary Movie 1, and Supplementary Fig. S1). When the extract solution was first added to surface-bound chromosomes, a small number of microtubules in the extract nucleated de novo, i.e., independent of chromosomes, and were visible at low density throughout the imaging field regardless of whether a chromosome was nearby (Supplementary Fig. S2a). When a de novo microtubule randomly happened to reach the vicinity of a chromosome, microtubules started to nucleate from it. These newly nucleated microtubules then served as seeds that nucleated more microtubules along them, leading to the formation of a microtubule network of uniform polarity (Fig. 1b, Supplementary Figs. S1 and S2b, and Supplementary Movie 1), similar to the networks observed in previous studies of branching microtubule nucleation[26,39]. In total, 50% of these early nucleation events occurred within 4 μm of a kinetochore and 55% of these nucleated microtubules grew at an angle less than 40° with respect to the kinetochore (Fig. 1c). These results demonstrate that most microtubule nucleation in this system is microtubule-dependent, i.e., it requires a de novo microtubule template to catalyze the formation of the eventual network. Furthermore, microtubule nucleation is amplified near and directed toward the kinetochores of isolated mitotic chromosomes.

### Branching microtubule nucleation is the principal source of chromosomal microtubules

Is branching microtubule nucleation indeed the principal source of chromosomal microtubule nucleation observed in Fig. 1? In order to assess this question, we individually immunodepleted the essential *Xenopus* branching factors TPX2 and augmin from the extract[26] and performed the TIRFM assay (Fig. 2a). In control experiments, using a random IgG antibody for immunodepletion, 34 ± 26% of chromosome clusters generated microtubule networks (n = 27 fields totaling 213 clusters) (Fig. 2a, left). In contrast, we rarely saw microtubule networks generated at chromosomes after depletion of either augmin or TPX2 (Fig. 2a middle and right, Supplementary Fig. S3, and Supplementary Movies 2 and 3). Less than 1 ± 2% of chromosome clusters for either augmin depletion or for TPX2 depletion generated microtubule networks (n > 10 fields per condition, with >135 clusters per condition) (Fig. 2b). These results suggest that branching microtubule nucleation, mediated by augmin and TPX2, is the key pathway to generate microtubules from chromosomes in *Xenopus* egg extracts. Therefore, in meiotic *Xenopus* egg extract spindles, chromosomal microtubule nucleation and branching microtubule nucleation are one and the same, and are the origin of the majority of early spindle microtubules.

### Model of branching microtubule nucleation in a uniform field of SAFs

Is branching microtubule nucleation sufficient to rationalize the organization of these chromosomal microtubule networks? To tackle this question, we turn to mathematical modeling. To establish a starting point for our model, we considered the simpler case of branching nucleation in a uniform field of SAFs. This condition can be probed experimentally by flowing *Xenopus* egg extract supplemented with the non-hydrolyzable Ran mutant RanQ69L into a microscope flow chamber, which creates a uniform field of SAFs where microtubules can branch (Fig. 3a and Supplementary Movie 4). We first observe the nucleation of a de novo mother microtubule that then

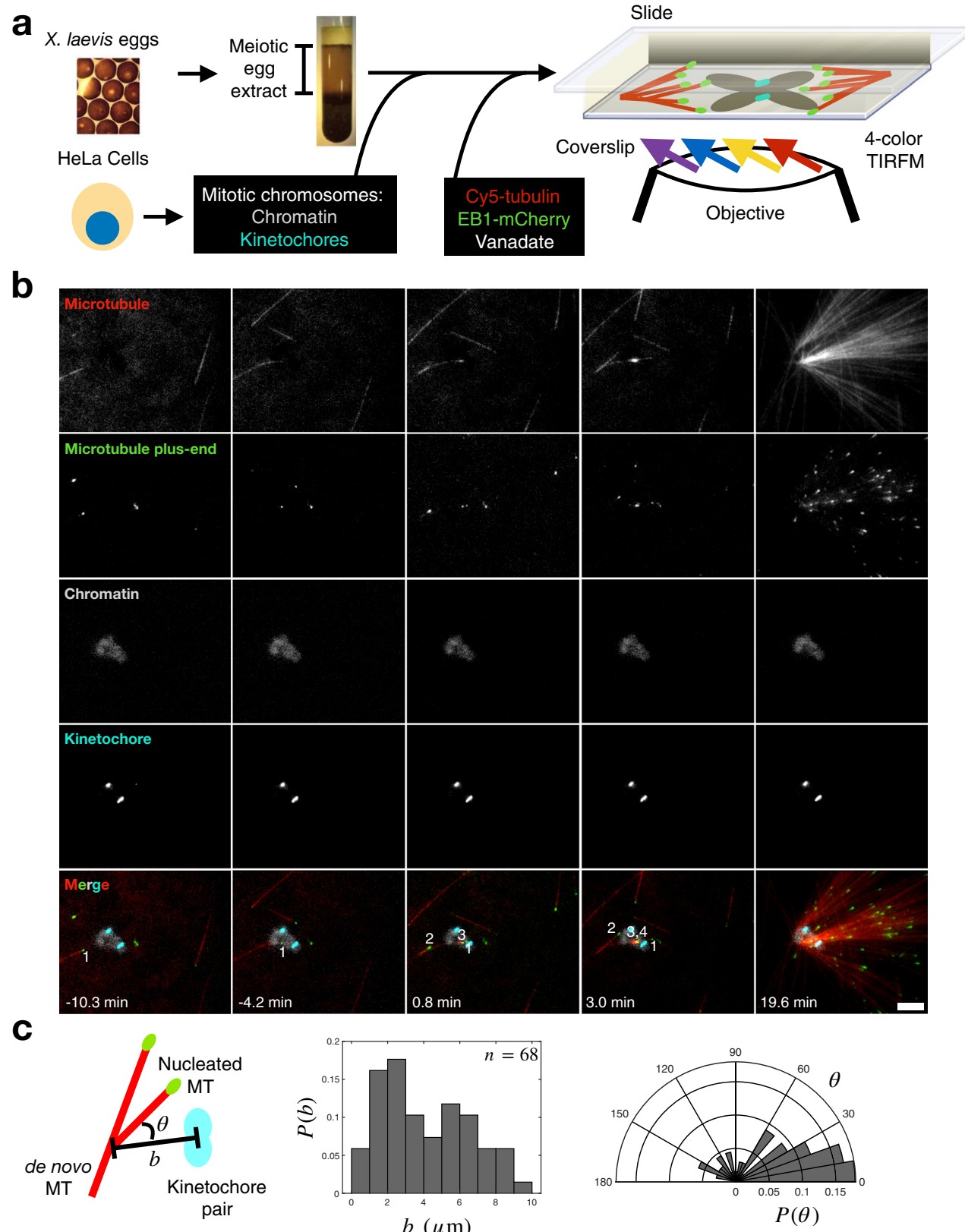

seeds the formation of a polar-branched network (Supplementary Movie 4).

To quantify the organization of these branched networks, we measured the spatial distribution of microtubule plus-end positions projected along the axis of the mother microtubule $x$ at different times $t$, where $t = 0$ corresponds to nucleation of the first branch (Fig. 3b, "Methods"). We measure the spatial distribution of plus-ends, as opposed to total tubulin density, to distinguish between the nucleation of new microtubules and the polymerization of existing ones. We normalized each plus-end distribution by its maximum values and rescaled its length by the maximum length of the final branched network. We averaged over $n = 10$ networks to obtain the final result

**Fig. 1 | Ex vivo reconstitution of microtubule-dependent microtubule nucleation near chromosomes. a** An illustration of the ex vivo reconstitution, which utilizes meiotic cytosol purified from *Xenopus laevis* eggs and chromosomes with CENPA-GFP labeled kinetochores purified from mitotic HeLa cells ("Methods"). Fluorescent tubulin and EB1 are included to label microtubules and microtubule plus-ends, respectively. Vanadate is used to inhibit motor activity. **b** The initial nucleation events near chromosomes are microtubule-dependent. A de novo microtubule randomly approaches a chromosome, allowing new microtubules to nucleate from it. This leads to an autocatalytic microtubule network of uniform polarity. Numbers demarcate unique microtubule plus-ends. $t = 0$ min corresponds to the first nucleation event. Scale bar is 5 μm. **c** Histogram of distance to the nearest kinetochore pair and polar histogram of angle towards the nearest kinetochore pair for up to the first ten nucleation events around chromosomes. Data are from 11 chromosome clusters across 7 extract preparations. $n = 68$ nucleation events.

(Fig. 3c, left). These branched networks grow autocatalytically for the first ~10 min after the addition of extract, consistent with previous work in *Xenopus* egg extract[39], and then saturate to a stationary state of constant average microtubule number (Fig. 3c, right).

To rationalize the structure of these branched networks, we developed a one-dimensional mathematical model of branching nucleation (Supplementary Information). Briefly, the model posits that microtubules can nucleate a branch with a probability proportional to the concentration of bound SAFs. In the limit where the unbound SAF concentration is uniform, our model admits an exact solution for the dimensionless plus-end distribution $\Phi$ as a function of dimensionless position $X$ and dimensionless time $T$,

$$\Phi(T,X) = \frac{1}{2\pi i} \int_{-i}^{i} ds \frac{1}{s} e^{sT + \left[\frac{B}{s+1} - (s+1)\right]X}, \qquad (1)$$

which is parameterized only by the "branching number" $B = K c_0 U / f_c$, where $K$ is a binding constant of SAFs to microtubules, $c_0$ is the SAF concentration, $U$ is the polymerization speed, and $f_c$ is the catastrophe frequency. The number of microtubules is given by $N(T) = \int_0^\infty dX \Phi(T,X)$. All dimensionless quantities are rescaled by measuring length in units of the average microtubule length $\langle \ell \rangle$ and measuring time in units of $\langle \ell \rangle / U$, where we use the typical *Xenopus* extract values of $\langle \ell \rangle = 8$ μm[28] and $U = 8$ μm/min[39] throughout the paper. Therefore, $B$ represents the competition between microtubule nucleation due to branching and microtubule turnover due to catastrophes.

For $B < 1$, microtubule turnover outcompetes branching nucleation, so microtubule plus-ends can only propagate a finite distance along the mother microtubule (Fig. 3d). After sufficient time, the statistically stationary distribution $\Phi(T \to \infty, X) = e^{(B-1)X}$ is achieved (Supplementary Information). When $B = 1$, microtubule turnover balances branching nucleation perfectly, and a branched network of constant microtubule density can propagate indefinitely (Fig. 3e). These constant density waves have been observed for large growing asters in interphase *Xenopus* extracts[40], where the authors offer a similar physical interpretation and present numerical results. By deriving an explicit formula, we see that their system corresponds to Eq. (1) when $B = 1$. If $B > 1$, branching microtubule nucleation outcompetes microtubule turnover, and an autocatalytic branched network forms that can propagate indefinitely (Fig. 3f). We find that our experimentally measured plus-end distribution (Fig. 3c, left) is well described by our theory with $B = 1.7$ (Fig. 3c, middle) during the autocatalytic growth phase.

After autocatalytic growth, the plus-end distribution reaches a statistically stationary state that is still consistent with $B > 1$ organization. If saturation were due to increasing microtubule turnover, the organization of the plus-end distribution would have to change to that predicted by $B < 1$, which we do not observe experimentally. Therefore, we attribute the reason for network saturation as owing to a limited pool of nucleating factors, and not to increasing microtubule turnover.

## Model of branching microtubule nucleation near chromosomes

To explain how chromosomes might generate branched networks, we modified our theoretical model to include the RanGTP pathway. Chromosomes release RanGTP at a flux $J$ into the extract, where it can be hydrolyzed into its inactive RanGDP form at a rate $k_H$, or it can bind to importin molecules that sequester SAFs, allowing SAFs to promote microtubule branching nucleation (Fig. 4a). This generates a concentration gradient of free SAFs centered at chromosomes. The simplest way to incorporate this pathway into our theoretical framework is to make the change $c_0 \to c_u(x) = c_0 \exp\left(-\frac{|x-d|}{\lambda}\right)$, where $\lambda$ is the length scale of the resultant exponential SAF gradient and $d$ is the distance between the initial de novo nucleation event and the chromosome (Fig. 4a and Supplementary Information). In applying our one-dimensional model to the experiments, we are assuming the de novo mother microtubule points toward the center of the chromosome, when in reality it is offset by an angle. However, this positive angle is always small in our experiments ($13 \pm 11°$, $n = 11$). Moreover, the gradient biases the branched network to point towards its center, since nucleation events are proportional to the SAF concentration. The initial de novo microtubule just needs to get close enough, after which the gradient will generate a polar-branched network directed at the chromosome. This idea both justifies the use of a one-dimensional model and helps rationalize why the initial branches are directed toward chromosomes (Fig. 1c).

In addition to the branching number $B$, the geometric ratio $\Lambda = \lambda / \langle \ell \rangle$ is now also a model parameter. To directly measure $\lambda$, we immunodepleted the SAF TPX2 from the extract and added back purified GFP-TPX2 at a physiological concentration of 100 nM[41]. We then proceeded with our TIRFM assay using this modified extract, observing GFP-TPX2 enrichment near the chromosomes (Fig. 4b, left). By plotting the GFP-TPX2 intensity as a function of distance from chromosomes, we see the emergence of an exponential gradient. We find a decaying exponential of length scale $\lambda = 23 \pm 2$ μm best fits our data ($n = 7$ chromosomes, $R^2 = 0.92$) (Fig. 4b, right), which sets $\Lambda = 3$ since $\langle \ell \rangle = 8$ μm[28], in excellent agreement with previous measurements[28]. We note that $\lambda$ is not appreciably affected by the number of microtubules nucleated in the vicinity of chromosomes, which reassures us that the dominant feature we are measuring is the concentration field of free, unbound TPX2 (Supplementary Fig. S4). This is the case because TPX2 is initially bound by importins and is uniformly soluble in a sequestered state. Near chromosomes, RanGTP will release TPX2, and because TPX2 can self-associate and condense[42], it will tend to form bright puncta when freed, which is what we observe.

When $\Lambda$ is finite, the model no longer admits a closed-form solution, so we solve numerically for the plus-end distribution (Supplementary Information). We find that the experimental (Fig. 4c, left) and theoretical (Fig. 4c, middle) distributions agree qualitatively with each other with $B = 2$. We see that a telltale sign of branching nucleation in a SAF gradient is that the plus-end distribution peaks downstream of the chromosomes ($x > d$); there is comparatively little plus-end density upstream of the chromosomes ($x > d$). Because both our experiments and theory have this feature, we attribute this asymmetry of the plus-end distribution to the spatial symmetry breaking caused by the SAF gradient. This asymmetry is intuitive: because we expect nucleation activity to be highest at the peak of the SAF gradient ($x = d$), and since the average microtubule has a length of $\langle \ell \rangle = 8$ μm, the resultant average plus-end distribution should be shifted by roughly this amount, which we observe (Fig. 4c, left). Moreover, because the

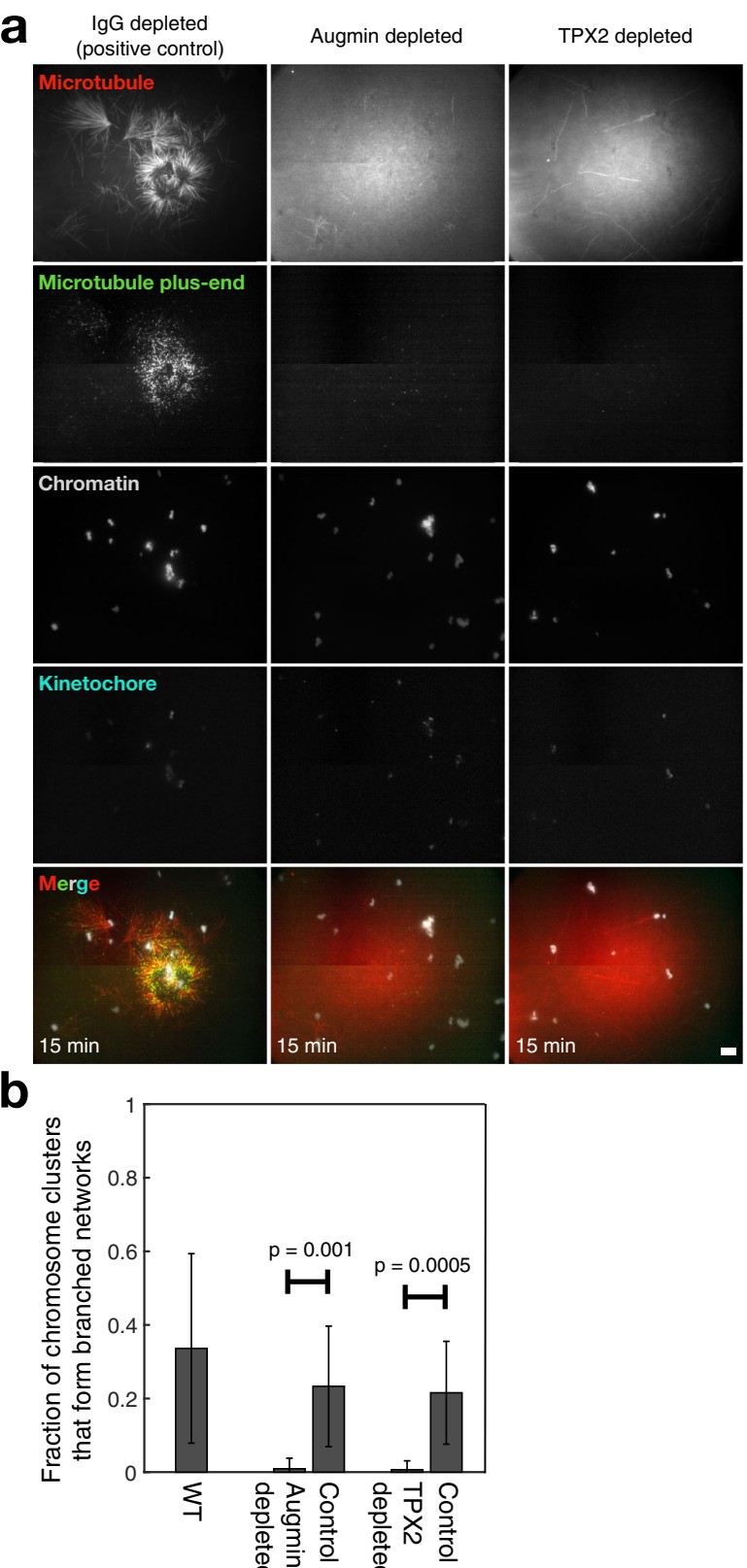

**Fig. 2 | Branching microtubule nucleation mediated by TPX2 and augmin is the chief source of chromosomal microtubule networks. a** Field of view for each of the three conditions tested: IgG depleted, augmin depleted, and TPX2 depleted, visualized using TIRFM in the presence of vanadate. Chromosomal microtubule networks were not generated when augmin or TPX2 was depleted. Scale bars are 10 μm. **b** Mean fraction of chromosomes that form microtubule networks in each condition. *n* = 213 chromosome clusters for wild-type extract, *n* = 135 for the augmin ID, *n* = 237 for the respective IgG control, *n* = 144 for the TPX2 ID, and *n* = 185 for the respective IgG control. Error bars are standard deviations across *n* = 10 imaging fields per condition. Data are from 4 extract preparations per condition. *P* values reported are from the two-sample Kolmogorov–Smirnov test.

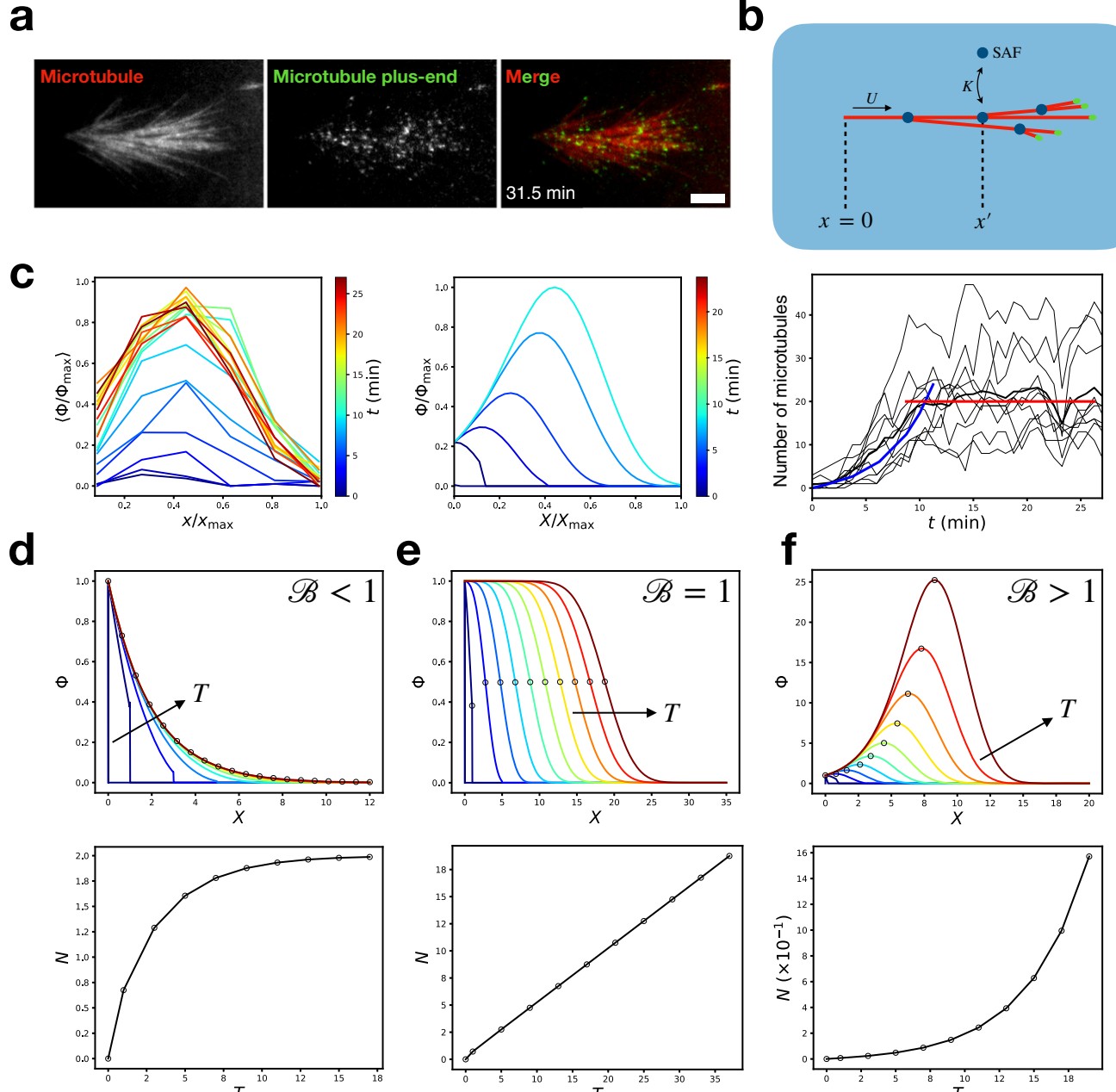

**Fig. 3 | Branching microtubule nucleation in a uniform field of SAFs: experiment and theory. a** TIRFM snapshot of a branched network in a uniform field of SAFs in the presence of vanadate. Scale bar is 5 μm. **b** Schematic of the branching nucleation model. **c** Experimentally measured plus-end distribution Φ (left panel, averaged over $n = 10$ branched networks across five different extract preparations) compared with the theoretically predicted plus-end distribution using $B = 1.7$ (middle panel). Plus-end distributions are normalized by their maximum value $\Phi_{max}$ and rescaled by the length of the final branched network $X_{max}$. In the right panel, the black curves show the experimentally measured number of microtubules, which increases exponentially over the first ~10 min before saturating. The blue curve is the theoretical prediction with $B = 1.7$ ($R^2 = 0.90$), while the red curve shows the average number of microtubules after saturation. **d** For $B < 1$, the plus-end distribution reaches a bounded stationary state set by microtubule turnover. **e** For $B = 1$, the plus-end distribution propagates as a constant density wave where microtubule turnover perfectly balances branching nucleation. **f** For $B > 1$, the plus-end distribution propagates as an autocatalytic growing front as branching nucleation outcompetes microtubule turnover.

SAF gradient is finite in extent, as opposed to the uniform field model, there is now a limiting pool of nucleating factors bounded by $\int_{-\infty}^{\infty} dx\, c_u(x)$, and thus our theory naturally captures both the early time amplification and late time saturation of the total number of microtubules versus time (Fig. 4b, right). Thus, we conclude that the observed organization of chromosomal microtubules in our system is consistent with a model of branching microtubule nucleation in an effector gradient spatially regulated by chromosomes.

## Reconstituting acentrosomal spindle assembly in *Xenopus* egg extract

Having determined where, when, and how microtubules are made at chromosomes in the absence of motor activity, we investigated what happens when motor activity is not inhibited in our TIRFM assay. As before, we initially observe de novo microtubules nucleating throughout the imaging field independent of chromosomes, although they are now mobile on the coverslip due to motor activity (Supplementary Movie 5). When one of these de novo microtubules enters the

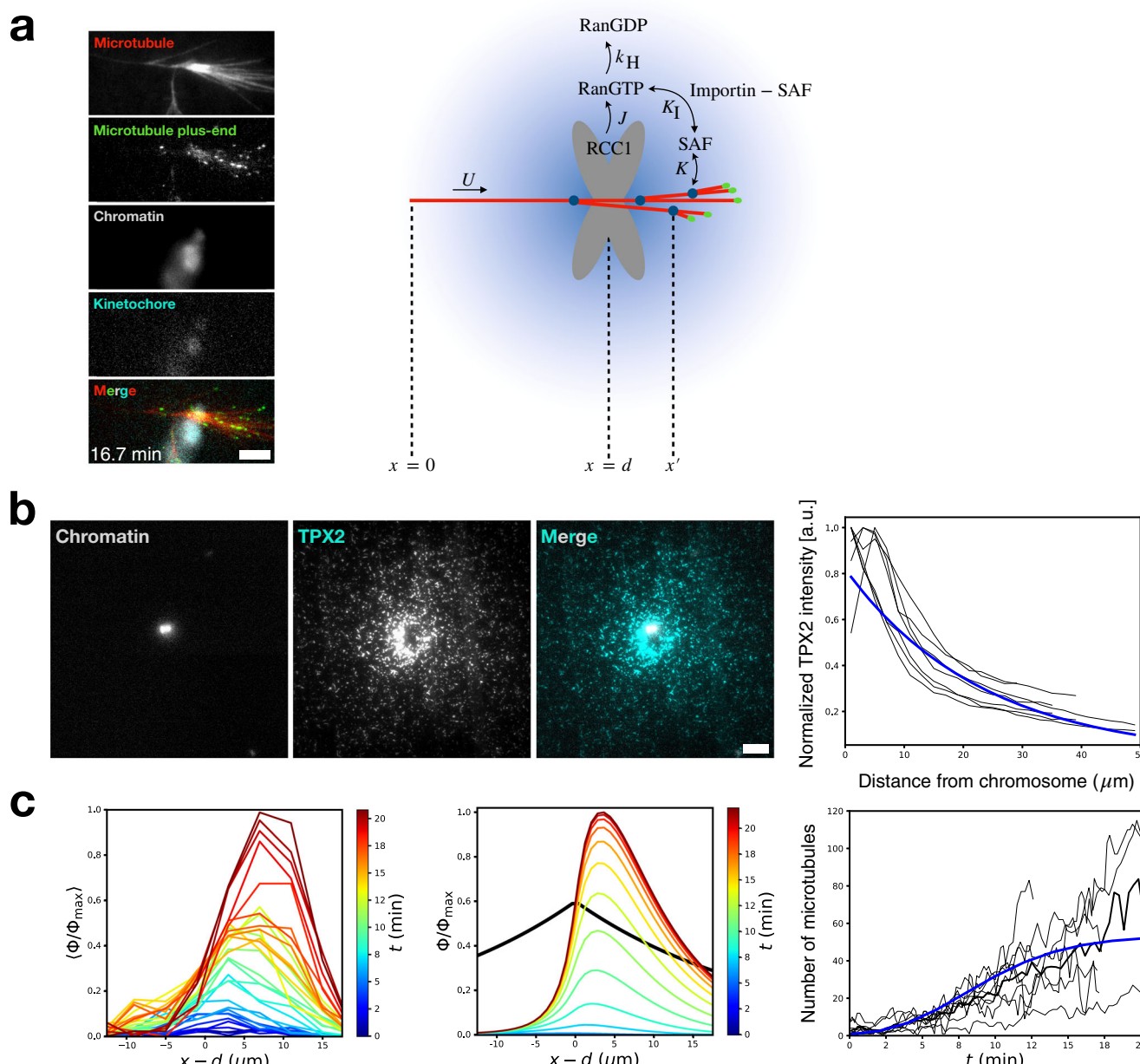

**Fig. 4 | Branching microtubule nucleation near chromosomes: experiment and theory. a** Representative TIRFM snapshot of a chromosomal branched network in the presence of vanadate and model schematic of RanGTP-regulated chromosomal branching nucleation. RanGTP is released at chromosomes and can either be hydrolyzed into RanGDP or bind the importin molecules that sequester SAFs, freeing them to promote microtubule branching nucleation. $t = 0$ min corresponds to the first branching event. Scale bar is 5 μm. **b** The images show that the SAF GFP-TPX2 is enriched around chromosomes in an extract depleted of endogenous TPX2. The plot shows intensity profiles of GFP-TPX2 as a function from the distance from chromosomes, computed using 2 μm radial bins (black curves). The blue curve is the average best fit decaying exponential $Ae^{-r/\lambda}$ with $A = 0.8 \pm 0.1$ and $\lambda = 23 \pm 2$ μm (7 chromosome clusters across 2 extract preparations). Scale bar is 5 μm. **c** Experimentally measured plus-end distribution $\Phi$ (left panel, $n = 11$ branched networks across 7 extract preparations) compared with the theoretically predicted plus-end distribution using $B = 2$ and $\Lambda = 3$ (middle panel). The black curve shows the SAF gradient profile. Plus-end distributions are normalized by their maximum value $\Phi_{max}$ and plotted as a function of distance from the chromosome $x-d$. In the right panel, the experimentally measured number of microtubules (black curves) is compared with the theoretical prediction using $B = 2$ and $\Lambda = 3$ (blue curve, $R^2 = 0.95$).

vicinity of a chromosome, branches nucleate from it and begin to grow near and toward kinetochores (Fig. 5a and Supplementary Movie 5). Motor activity then reorganizes these branched networks to produce multipolar microtubule networks, i.e., spindles (Fig. 5b and Supplementary Movies 5 and 6). The spindles that assemble in the presence of motor activity (Fig. 5b) stand in sharp contrast to the uniformly polar-branched networks seen in Figs. 1–4.

To further quantify the results of the TIRFM assay in the presence of motors (Fig. 5c and Supplementary Movie 6), we measured the total

tubulin intensity and number of EB1 spots over time in a 40 μm × 40 μm box around chromosomes. We found that the total microtubule mass and number of microtubules in these spindles plateau ~10 min after the onset of spindle assembly (Fig. 5d), consistent with our experiments using vanadate to inhibit motors (Fig. 3c, right and Fig. 4c, right). The time to the plateau is also consistent with previous findings based on similar experiments with chromatin beads[37,43]. Because both the number of microtubules and total microtubule mass plateau around the same time (Fig. 5d), this further supports the idea that

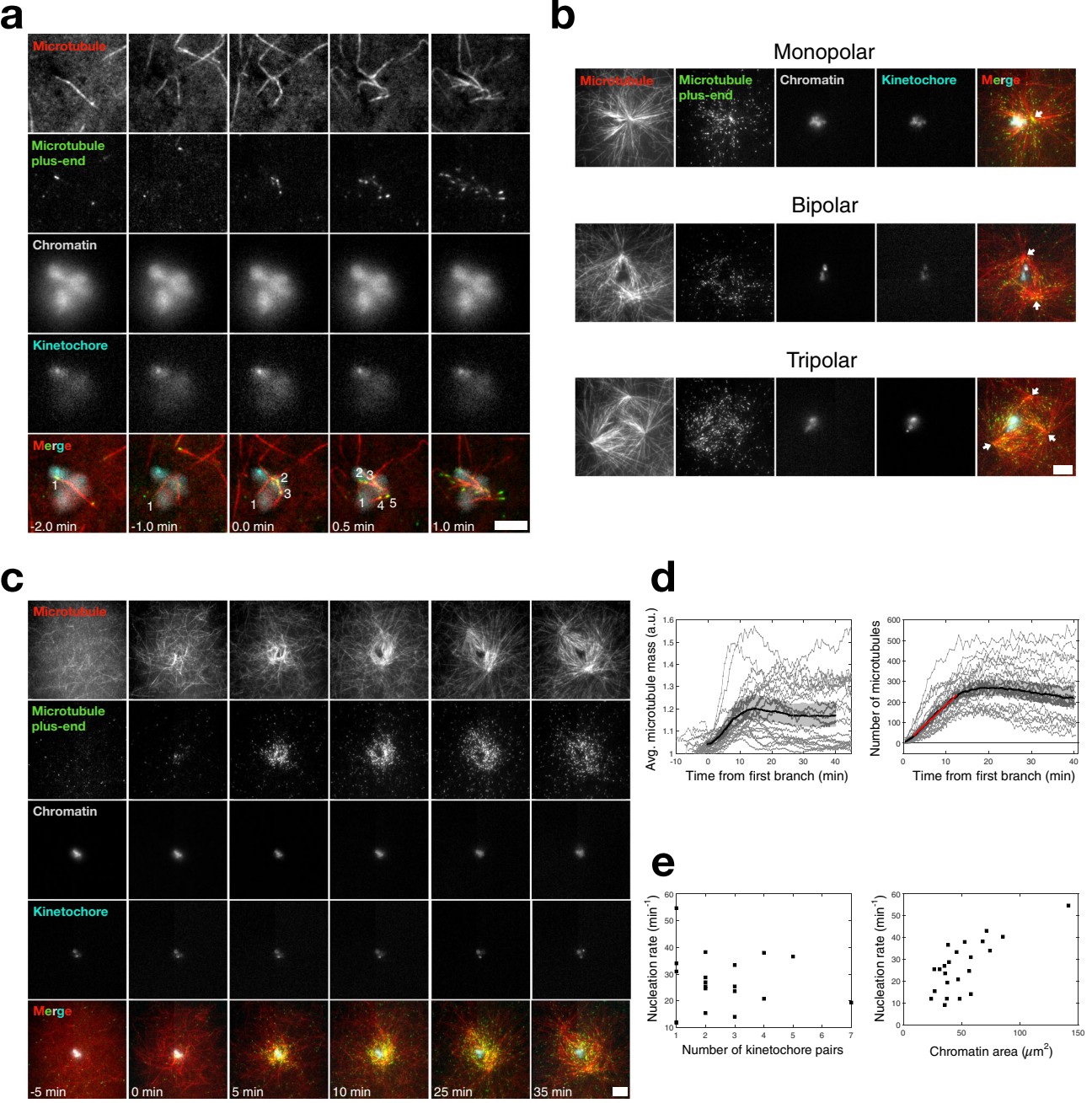

**Fig. 5 | In the presence of motor activity, chromosomes generate polar spindles. a** Branching microtubule nucleation leads to the formation of branched networks at chromosomes, which are eventually organized into a spindle by motor activity. A de novo microtubule randomly approaches a chromosome, bends due to motor activity, and nucleates new microtubule branches. This leads to an auto-catalytic branched microtubule network that gets reorganized by motors into a bipolar spindle over time. Numbers demarcate unique microtubule plus-ends. $t = 0$ min corresponds to the first branching event. Scale bars are 5 μm. **b** TIRFM visualization of mono-, bi-, and tripolar spindles assembled around purified chromosome clusters on the coverslip surface. White arrows mark the poles. Scale bars is 10 μm. **c** Time-lapse TIRFM images of a bipolar spindle assembling around chromosomes. $t = 0$ min corresponds to the first branching event. Scale bar is 10 μm. **d** Microtubule mass (left) and number of microtubules (right) versus time during spindle assembly. Microtubule mass and number plateau at -10 min. $n = 23$ spindles across seven extract preparations. Shaded regions represent 95% bootstrap confidence intervals. The red curve is a linear fit, giving an effective microtubule nucleation rate of $k = 19.5 \pm 0.54$ microtubules/min (mean ± 95% confidence bounds, $R^2 > 0.99$). **e** Nucleation rate in the spindle does not significantly correlate with the number of visible kinetochore pairs in the chromosome cluster (left). One-sided Pearson correlation coefficient = −0.07, $P = 0.78$. $n = 19$ spindles across 7 extract preparations. The nucleation rate in the spindle correlates with the two-dimensional projected area of the chromatin in the chromosome cluster (right). One-sided Pearson correlation coefficient = 0.73, $P = 0.0003$. $n = 23$ spindles across 7 extract preparations.

microtubule mass in the spindle is limited by available nucleators, and not by tubulin availability or changing microtubule polymerization dynamics.

We next asked if acentrosomal spindle assembly relies on the number of chromosomes or the amount of chromatin. To determine the number of chromosomes, we counted the number of kinetochore pairs, although we note that there could be more kinetochores above the focal plane of TIRFM. We found that there was no significant correlation between nucleation rate and the number of chromosomes, which we counted in each chromosome cluster (Fig. 5e, left; Pearson

correlation coefficient = −0.07, $P = 0.78$). In contrast, there was a positive and significant correlation between the nucleation rate and the in-plane area of the chromatin in each chromosome cluster (Fig. 5e, right; Pearson correlation coefficient = 0.73, $P < 0.001$). Thus, these results show that the amount of chromatin in chromosomes sets the nucleation rate of microtubules in the spindle, independent of the number of chromosomes. This is consistent with our theoretical model, since the branching number $B$ scales with the SAF concentration $c_0$, which in turn scales with the flux of RanGTP $J$ produced by chromosomes. Since $J$ is proportional to the total surface area of chromatin, we expect the nucleation rate to be proportional to the total chromatin area.

Lastly, we investigated whether microtubules polymerized directly through chromosomes. To test this idea, we let acentrosomal spindles form around chromosomes for 20 min and then plotted the mean intensity of the tubulin channel and all microtubule tracks around chromosomes (Supplementary Fig. S5). Our observed microtubule tracks suggest that microtubules can only pass through less compact regions of chromosomes, such as the chromosome arms. In stark contrast, the tubulin intensity and microtubule tracks both featured voids at the kinetochores. No microtubules, out of thousands in a spindle, passed through the kinetochores of a chromosome—even when earlier growth was oriented toward a kinetochore. These results suggest that microtubules cannot polymerize through kinetochores in this system.

### Analysis of bulk spindle organization in *Xenopus* egg extract

Because the purified chromosomes are immobilized to the coverslip, they are prevented from rearranging to accommodate the forces from motors, leading to spindle poles that are less stable (Fig. 5b and Supplementary Movie 6). By imaging in the TIRF field, we additionally cannot observe poles that might form above the coverslip. Also, our TIRFM assays were performed using DAPI-stained chromosomes, which might interfere with DNA resulting in less robust spindles. To overcome these issues, we incubated unstained chromosomes in bulk extract and allowed spindles to form in a three-dimensional environment. After 45 min, samples were diluted and spin-mounted onto coverslips, after which they were chemically fixed and immunostained for proteins involved in spindle self-organization and for chromatin. Each structure was imaged as a z-stack using epifluorescence microscopy ("Methods"). Just as in the TIRFM assay, we observed multipolar structures organized around chromosomes (Fig. 6a, b), where the bipolar spindles now display a robust ellipsoidal shape (Supplementary Fig. S6a). We immunostained for NuMA (Nuclear Mitotic Apparatus), the adapter for cytoplasmic dynein that allows it to bind microtubules and cluster them at their minus ends, forming spindle poles[44]. As expected, NuMA displayed sharp polar localization (Fig. 6a and Supplementary Fig. S6a), allowing us to unambiguously identify and count functional poles. We also stained for Eg5 (kinesin-5), the tetrameric motor that slides antiparallel microtubules apart to promote poleward flux[45,46]. We found that Eg5 localized to the entire spindle with enrichment at the poles (Fig. 6b), in agreement with previous measurements on *Xenopus* egg extract metaphase spindles[47]. A negative control random IgG showed no significant or distinct localization to the spindle (Supplementary Fig. S6b). These results confirm that our reconstituted system can form proper spindles with the correct motor localizations.

Over 50% of all imaged spindles ($n = 173$) were properly bipolar (Fig. 6c). Because our reconstituted system can generate a wide distribution of chromosome cluster sizes with labeled kinetochores, we asked if the resultant spindle polarity scaled with the chromatin area or number of chromosomes. We found that both increasing chromatin area and the number of chromosomes lead to an increasing number of poles (Fig. 6d, e). This is consistent with a recent study that showed that increasing amounts of branching nucleation can stabilize an additional spontaneously formed pole[48]. Interestingly, single chromosomes were only capable of generating monopoles (Fig. 6e), suggesting that there is a minimum number of chromosomes required for the organization of a proper bipolar spindle.

## Discussion

In this work, we report experimental and theoretical advances to investigate acentrosomal spindle assembly in *Xenopus laevis* egg extracts, and can now propose the following model (Fig. 6f). First, de novo microtubules nucleate randomly throughout the cytoplasm. When one of the microtubules finds its way near chromosomes, branched microtubules nucleate from it due to the action of the RanGTP-mediated SAF gradient around chromosomes. In the absence of motors, branched microtubule networks of uniform polarity directed toward chromosomes form (Figs. 1 and 4). When motors are present, self-organized multipolar microtubule networks form around chromosomes (Figs. 5 and 6).

It is important to discuss other systems where the mechanism of acentrosomal spindle assembly is different from our model. For example, previous work in mouse oocytes showed that multiple small microtubule organizing centers (MTOCs) with pericentrin-rich poles consisting of many microtubules coalesce and seed the subsequent Ran-dependent assembly of spindle mass after the breakdown of the nuclear envelope[49]. Whether chromosomes contribute to this Ran-dependent assembly, by releasing factors that enable branching nucleation, remains to be determined. In meiosis I, as opposed to the meiosis II arrested extract we study here, early spindle assembly initiates near the disassembling nuclear envelope in *C. elegans*[50] and is independent of RanGTP in mice and *Xenopus* oocytes[51]. Thus, acentrosomal spindle assembly in meiosis I is biophysically distinct from that in meiosis II.

Previous work suggested that microtubules formed at random locations and orientations at the start of meiotic spindle assembly, based on imaging techniques that could not resolve individual filaments[6,37,45]. In contrast, because TIRFM provides excellent signal-to-noise ratio, we see that single de novo microtubules seed the formation of branched microtubules that are no longer randomly oriented, but rather nucleate near and are directed towards chromosomes, which we attribute to the spatial bias generated by the SAF gradient. Consequently, the capture of chromosomes by the spindle during meiosis in *Xenopus* is made more likely.

How kinetochores are efficiently captured and kinetochore fibers form in both centrosomal and acentrosomal systems is a key question in cell biology. Our results show one pathway for acentrosomal spindle assembly uses polarized, branched networks that point toward chromosomes due to a spatially biasing SAF gradient. A recent study[27] lends support to this idea, as it was found in human mitotic cells that microtubules nucleated towards kinetochores in an augmin-dependent manner, and that this process is essential for kinetochore fiber maintenance during metaphase. We note that our model does not require nor suggest that kinetochores themselves specifically stimulate branching nucleation, but rather relies only on the ability of chromatin-bound RCC1 to produce RanGTP to release SAFs in a gradient around chromosomes. This contrasts with other proposed mechanisms whereby kinetochores themselves can either recruit γTuRC directly via the Nup107–160 nucleoporin[14] or recruit pericentrin to their transient fibrous corona[52], both of which allow for microtubule nucleation directly from kinetochores. It might be that these pathways play a supporting role in our system. To know for certain would require performing additional TIRFM assays with chromosomes lacking kinetochores and isolated kinetochores and comparing the resulting microtubule networks.

In our system, branching microtubule nucleation is spatially regulated by a RanGTP-mediated SAF gradient (Fig. 4). Indeed, we explicitly showed that this was the case for TPX2 (Fig. 4b), a known SAF

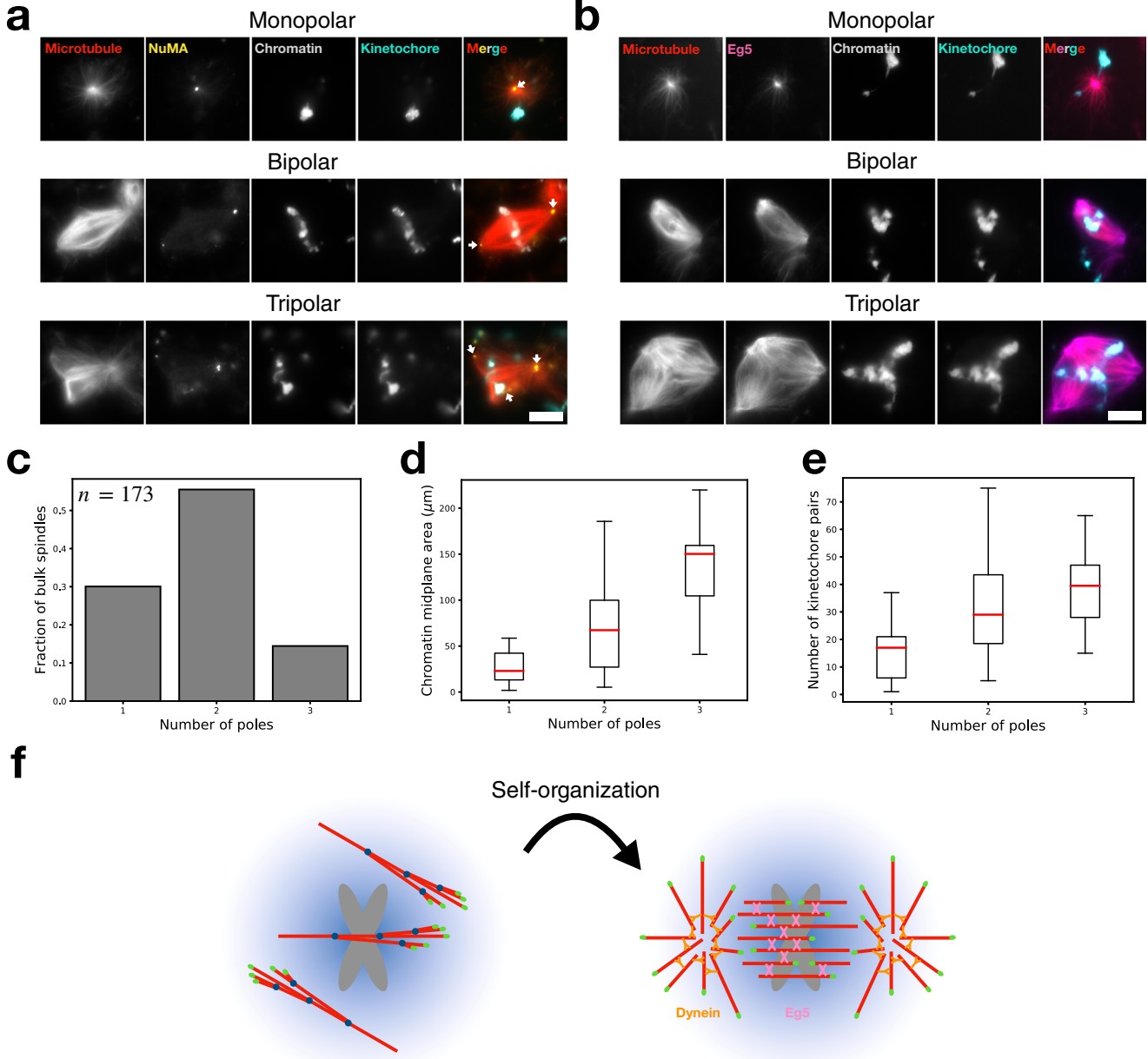

**Fig. 6 | Immunofluorescence and analysis of bulk spindle organization.** Epifluorescence visualization of fixed mono-, bi-, and tripolar spindles organized around purified chromosome clusters in bulk extract immunostained for **a** NuMA and **b** Eg5. Images shown are midplanes of a z-stack. White arrows mark the poles which show sharp NuMA localization. Scale bars are 10 μm. **c** Number of monopolar (*n* = 52), bipolar (*n* = 96), and tripolar (*n* = 25) spindles. *n* = 173 spindles total across 13 extract preparations. The number of spindle poles increases with **d** increasing chromatin area and **e** increasing chromosome number. Chromatin areas were measured at the spindle midplane. Chromosome numbers were manually determined by counting kinetochore pairs throughout each z-stack. The minimum number of chromosomes that formed a monopolar, bipolar, or tripolar spindle were 1, 5, and 15, respectively. Boxes are interquartile ranges, whiskers extend 1.5× the interquartile range, and red lines are medians. **f** Schematic of our general model for acentrosomal spindle assembly. First, de novo microtubules randomly enter the SAF gradient generated by chromosomes. Branching microtubule nucleation occurs along these first mother microtubules, generating branched networks. These branched networks are then self-organized by molecular motors such as dynein and Eg5 into a bipolar spindle.

and essential branching factor in *Xenopus*[53]. However, in other systems such as *Drosophila* oocytes[54] and human cells[55], TPX2 is not required and instead augmin is the only necessary branching factor along with γ-TuRC. Thus, for our model to apply across a wider range of systems, augmin would also need to be spatially regulated by importins and RanGTP. Excitingly, two recent studies have reported that augmin is indeed regulated by RanGTP[56,57], extending the applicability of our model.

Because we used purified chromosomes, we were able to observe that no microtubule tracks pass through the kinetochores in the center

of chromosomes, yet tracks do appear to pass through the peripheral chromatin arms (Supplementary Fig. S5). This finding suggests that microtubules, which are 25 nm in diameter, may pass through gaps within chromatin. It is known that microtubules interact with other proteins on the surface of chromosomes besides kinetochores, such as chromokinesin motors to generate polar ejection forces that move chromosomes away from the spindle poles[58–62]. Considering our data, the possibility that microtubules interact with proteins or DNA within chromosomes suggests new avenues for investigation of the intersection of chromatin and cytoskeletal biology. For example, it has been

recently suggested that chromosomes can regulate their structure to prevent or enhance microtubules from perforating them[63].

There are several technical limitations of our study worth highlighting. For example, our mitotic chromosomes purified from HeLa cells might react in an unphysiological way with *Xenopus* meiotic egg extract, either by not remaining properly condensed or by binding antagonistic *Xenopus* proteins. This may help explain why only ~30% of our chromosome clusters nucleate branched microtubule networks (Fig. 2b). Furthermore, defects in chromosome structure may impact the ability of microtubules to penetrate the chromatin mesh. Another limitation comes from using vanadate to inhibit motors, which is a nonspecific ATPase inhibitor that may inhibit polymerases or severases that utilize ATP to tune microtubule polymerization. Since branching nucleation depends strongly on the amount of available microtubule mass, this would affect the number of microtubules generated. We also speculate that motors can pull out branched microtubules from their branch site, and thus enable a new round of branching microtubule nucleation to take place from the same site. These points may help explain the differences in microtubule numbers between Figs. 4c and 5d, but the precise role that motors may play in regulating branching nucleation remains an open question.

Based on the framework and mechanism we provide here for how branching microtubule nucleation in a SAF gradient around chromosomes occurs, the time is now ripe to further investigate how molecular motors reorganize these initial chromosomal branched networks into a functioning bipolar spindle. Such future studies would help further rationalize the novel scaling relationships we report in Fig. 6d, e. These issues are starting to be quantitatively addressed[64–66], with the ultimate goal being to minimally reconstitute a functional acentrosomal spindle in vitro.

## Methods

### Ethics
Animal care was done in accordance with recommendations in the Guide for the Care and Use of Laboratory Animals of the NIH and the approved Institutional Animal Care and Use Committee (IACUC) protocol 1941-16 of Princeton University.

### Protein expression and purification
EB1-mCherry was purified as previously described[67]. Protein was expressed in *E. coli* (strain Rosetta 2) for 4 h at 37 °C. Cells were lysed via a French press using an EmulsiFlex (Avestin) in lysis buffer (50 mM NaPO$_4$, pH 7.4, 500 mM NaCl, 20 mM imidazole, 2.5 mM PMSF, 6 mM CME, 1 cOmplete™ EDTA-free Protease Inhibitor (Sigma), 1000 U DNAse 1 (Sigma)). Protein was affinity purified from the lysate using a HisTrap HP 5 mL column (GE Healthcare) in binding buffer (50 mM NaPO$_4$, pH 7.4, 500 mM NaCl, 20 mM imidazole, 2.5 mM PMSF, 6 mM BME). Protein was then eluted using elution buffer (50 mM NaPO$_4$, pH 7.4, 500 mM NaCl, 500 mM imidazole, 2.5 mM PMSF, 6 mM BME). Next, peak fractions were pooled and loaded onto a Superdex 200 pg 16/600 gel filtration column, and gel filtration was done in CSF-XB (10 mM HEPES, pH 7.7, 1 mM MgCl$_2$, 100 mM KCl, 5 mM EGTA) with 10% (w/v) sucrose.

RanQ69L, used to test the quality of meiotic *Xenopus* egg extract prior to experiments with chromosomes, was also purified as previously described[67]. RanQ69L, tagged on its N-terminus with BFP to improve solubility, was expressed and then lysed as above in lysis buffer (100 mM tris-HCl, pH 8.0, 450 mM NaCl, 1 mM MgCl$_2$, 1 mM EDTA, 0.5 mM PMSF, 6 mM BME, 200 μM GTP, 1 cOmplete™ EDTA-free Protease Inhibitor, 1000 U DNAse 1). Protein was then affinity purified from the lysate using a StrepTrap HP 5 mL column (GE Healthcare) in binding buffer (100 mM tris-HCl, pH 8.0, 450 mM NaCl, 1 mM MgCl$_2$, 1 mM EDTA, 0.5 mM PMSF, 6 mM BME, 200 μM GTP). Bound protein was eluted using elution buffer (100 mM tris-HCl, pH 8.0, 450 mM NaCl, 1 mM MgCl$_2$, 1 mM EDTA, 0.5 mM PMSF, 6 mM BME, 200 μM GTP, 2.5 mM D-desthiobiotin). Finally, eluted protein was dialyzed into CSF-XB (10 mM HEPES, pH 7.7, 1 mM MgCl$_2$, 100 mM KCl, 5 mM EGTA) with 10% (w/v) sucrose overnight.

Tubulin from bovine brain (PurSolutions) was labeled with Cy5 NHS ester (GE Healthcare) as previously described[68].

Protein concentration was assessed using SDS-PAGE followed by Coomassie staining against a standard of BSA with known concentrations, or via Bradford dye (Bio-Rad).

### Chromosome isolation
Chromosomes were isolated from mitotic HeLa cells following previous approaches[69–72]. First, GFP-CENPA HeLa cells were synchronized to mitosis via a single or double thymidine block[73]. Prior to collecting cells, cytochalasin D was added to media at a final concentration of 10 mg/mL. Next, cells were collected and swelled in a hypotonic solution of 0.075 M KCl for 20 min at 37 °C. After this, all work was done at 4 °C. The cells were centrifuged at 780×$g$ for 15 min. The supernatant as removed and then 25 mL of polyamine solution (15 mM tris-HCl, pH 7.4, 2 mM EDTA, 80 mM KCl, 20 mM NaCl, 0.2 mM spermine, 0.5 mM spermidine, 0.05% (v/v) Empigen BB (Sigma), 7 mM BME, 1 cOmplete™ EDTA-free Protease Inhibitor) was layered over the pellet. The pellet and polyamine solution were kept on ice for 5 min and then gently resuspended. Cells were then gently lysed by a Dounce homogenizer (B pestle, 10 passes). The lysate was gently centrifuged at 190 × $g$ for 3 min, 7/10th of the supernatant taken, and then the supernatant was more strongly centrifuged at 1750×$g$ for 20 min onto a 70% (v/v) glycerol cushion of the polyamine solution. The layer of chromosomes above the cushion was gently resuspended with the cushion. Next, the resuspended cushion sample and 30 mL Percoll buffer (5 mM tris-HCl, pH 7.4, 20 mM EDTA, 20 mM KCl, 0.8 mM spermine, 2.25 mM spermidine, 1% (v/v) thiodiethanol, 0.05% (v/v) Empigen BB, 89% (v/v) Percoll (GE healthcare), 1 cOmplete™ EDTA-free Protease Inhibitor) were added to a Dounce homogenizer to a final volume of about 35 mL and gently homogenized (B pestle, 10 passes). Then, the homogenized solution was brought to 55 mL with more Percoll buffer and centrifuged at 48400 g for 30 min in a 45 Ti rotor (Beckman). Chromosomes appeared as a faint band about 1/5th from the bottom of the centrifuge tube. The Percoll gradient was manually fractionated and fractions were imaged for chromosomes via epifluorescence. Chromosome-rich fractions were pooled, diluted threefold in dilution buffer (5 mM tris-HCl, pH 7.4, 20 mM EDTA, 20 mM KCl, 0.8 mM spermine, 2.25 mM spermidine, 1% (v/v) thiodiethanol, 0.05% Empigen BB, 1 cOmplete™ EDTA-free Protease Inhibitor), and centrifuged at 1250 g for 20 min onto a 2 M sucrose CSF-XB (10 mM HEPES, pH 7.7, 1 mM MgCl$_2$, 100 mM KCl, 5 mM EGTA, 2 M sucrose) cushion twice, resuspending the first cushion and sample with CSF-XB (10 mM HEPES, pH 7.7, 1 mM MgCl$_2$, 100 mM KCl, 5 mM EGTA). The final cushion and sample were gently resuspended and then aliquoted and flash-frozen.

### Surface chemistry for chromosome attachment
Flow chambers were made using double-sided tape to create a rectangular chamber between a glass slide and a coverslip. Anti-ds DNA antibody (Abcam, ab27156) was flowed in at 0.1 to 0.17 mg/mL and allowed to adhere to coverslips for 10 min. Excess antibody was washed out three times using CSF-XB (10 mM HEPES, pH 7.7, 1 mM MgCl$_2$, 100 mM KCl, 5 mM EGTA) with 10% (w/v) sucrose, and then the coverslip surface was passivated using $\kappa$-casein at 1 mg/mL, incubated for 10 min. Next, diluted, purified chromosomes stained with DAPI (0.1–1 μg/mL) were flowed into the chamber and allowed to bind for 10 to 20 min. Unbound chromosomes were washed out three times using CSF-XB (10 mM HEPES, pH 7.7, 1 mM MgCl$_2$, 100 mM KCl, 5 mM EGTA) with 10% (w/v) sucrose. Meiotic *Xenopus* egg extract was flowed into the chamber and chromosomal microtubule nucleation was visualized using TIRFM.

Coverslips (no. 1.5, Fisher or equivalent) were silanized for immunodepletion and drug inhibition experiments following an existing protocol[68]. Briefly, coverslips were sonicated using a bath sonicator for 5 min in 1 M NaOH, rinsed with DI water twice, sonicated in DI water for 5 min, and then dried using inert nitrogen gas. Coverslips were then silanized for 1 h using a 0.05% (v/v) dichloro(dimethyl) silane (DDS) solution in trichloroethylene (TCE), with gentle stirring. Coverslips were then sonicated through a series of methanol baths for 5, 15, and 30 min, and finally dried using inert nitrogen gas. Silanized coverslips were stored in clean, sealed glass containers and used within 1 week.

## Meiotic *Xenopus laevis* egg extract purification, protein immunodepletion, and drug inhibition

Meiotic *Xenopus laevis* egg extract, also known as M-phase, metaphase arrested, or CSF extract, was prepared from *Xenopus laevis* eggs according to previously described protocols[74,75]. Egg extract was diluted no more than 75% for all TIRFM experiments and prepared as 20 μL reactions containing 15 μL extract, 0.89 μM Cy5-labeled tubulin, and 200 μM EB1-mCherry. CSF-XB (10 mM HEPES, pH 7.7, 1 mM $MgCl_2$, 100 mM KCl, 5 mM EGTA) with 10% (w/v) sucrose was added to bring final dilution down to 75%, as needed. The freshly prepared extract was tested for its ability to generate branched microtubule networks before further experiments were done or immunodepletion was started by visually comparing reactions without and with 10 μM BFP-RanQ69L. For depletion of TPX2 or augmin from egg extract, 72 μg of purified anti-TPX2 (in-house, Genscript) or anti-Haus1 (in-house, Genscript) antibody, as used previously[39], was coupled overnight to 300 μL protein A magnetic beads (Dynabeads, ThermoFisher). The following day, 150 μL fresh egg extract was depleted of either target protein in three rounds of washes with beads, using 100 μL of beads per wash, 20 min per round. Control depletions were done using the same amount of a random rabbit IgG antibody (Sigma, I5006). The efficiency of depletion using these antibodies was previously determined by western blot and confirmed by assaying for Ran aster generation without and with 10 μM BFP-RanQ69L. Protein and control-depleted extracts were imaged simultaneously using multichannel flow chambers made using multiple pieces of double-sided tape. For inhibiting motors using $Na_3VO_4$ (sodium orthovanadate or 'vanadate') (NEB), vanadate was added to extract at a final concentration of 0.5 mM.

*Xenopus laevis* husbandry was done in accordance with the recommendations in the Guide for the Care and Use of Laboratory Animals of the National Institutes of Health. All animals were cared for according to the approved Institutional Animal Care and Use Committee (IACUC) protocol 1941-16 of Princeton University.

## 4-color time-lapse TIRFM and image analysis

Total internal reflection fluorescence microscopy (TIRFM) and epifluorescence microscopy was performed using a Nikon TiE microscope with a 1.49 NA, ×100 magnification objective. An Andor Zyla sCMOS camera was used for acquisition, with software NIS elements (Nikon). For all imaging experiments, multiple fields were imaged in parallel to facilitate sampling more chromosomes. For depletion and drug inhibition experiments, multiple fields were imaged in parallel to enable sampling from experimental and control extract reactions at nearly the same times, and one channel free of chromosomes was imaged to confirm low background nucleation levels in the extract.

The polarity of spindles around chromosomes and the number of chromosomes with microtubule networks were determined by manual counting. Normalized tubulin intensity over time was determined by taking the average pixel intensity in a 40-μm × 40-μm window centered around the chromosome over time and dividing by the minimum average value. For measuring intensity over time for the depletion and drug inhibition experiments, a 10-μm × 10-μm window was used to avoid counting microtubules generated from adjacent chromosomes

in the same image field. The time of the initial branch was determined by visual inspection. The number of microtubules was counted, and microtubules were tracked using TrackMate v5.2.0 as implemented in Fiji (ImageJ)[76]. The accuracy of the tracking was checked visually.

Plus-end distributions were computed by binning EB1 comets from TrackMate into 4-μm bins along the axis of the polar-branched network set by the de novo mother microtubule for all acquired frames. Distributions were then normalized by their maximum values and averaged together to generate the final average distribution. For chromosomal branched networks, we only considered chromosomes that were sufficiently isolated so that the branched networks from nearby chromosomes could not interfere with the plus-end distribution measurement for a single chromosome cluster.

Chromatin area measurements were determined by thresholding chromatin images in MATLAB such that the intraclass variance between pixel sets was minimized (Otsu's method). The resulting binarized images were eroded and then dilated. Identical parameters were used for each step while thresholding chromatin images. The locations and angles of the earliest microtubule nucleation events were found manually using Fiji. The number of kinetochores and centromere void regions was determined by visual inspection of kinetochore and chromatin images and tubulin mean intensity time projections.

## Immunofluorescence imaging and analysis of fixed bulk spindles

For bulk spindle assembly assays, a reaction mixture of 17 μL of fresh *Xenopus* egg extract, 2 μL of 1-to-2 diluted chromosomes from our final purified aliquots, and 1 μL of 2 mg/ml Cy5-labeled tubulin was allowed to incubate at 16 °C for 45 min. Circular coverslips (Fisherbrand 12-545-80) were completely submerged in a 0.01% w/v poly-L-lysine solution for 10 min, then set to dry completely on lens paper before use. Spin-down tubes were prepared by placing a plastic adapter in the bottom of a 15 mL glass tube, then circular coverslips were placed on top of the adapter[75]. In total, 5 mL of spindle cushion (1× BRB80, 40% w/w glycerol) was gently pipetted over the coverslip. After incubation, the reaction mixture was diluted with 1 mL of spindle dilution buffer (1× BRB80, 30% w/w glycerol, 0.5% v/v Triton-X-100) using a cut pipette tip, mixed gently, and then gently pipetted on top of the spindle cushion. Samples were spun down at 10,200 RPM in a TH13-6 × 50 swinging bucket rotor for 15 min at 16 °C. Coverslips were then removed from the tube, ~200 μL of cold methanol was pipetted onto them, and then they were incubated at −20 °C for 5 min. Coverslips were then placed in a room temperature humidity chamber, washed with PBS-N (1× PBS, 0.1% v/v NP40), and blocked with PBS-B (1× PBS, 0.3% w/v BSA) for 20 min. The primary antibody, either anti-NuMA (Invitrogen, GT3511) or anti-Eg5 (custom, Genscript), was then incubated on the coverslip at 5 μg/ml for 45 min, washed with PBS-N, then the secondary antibody (Invitrogen, A-11011) was incubated at 1 μg/ml for 45 min, and then washed again with PBS-N. The coverslip was then incubated with a Hoechst solution (1× PBS-N, 5 μg/ml Hoechst) for 5 min, then washed with PBS-N. In total, 1 μL of mounting media (ProLong Diamond Antifade, P36965) was pipetted onto a glass slide, the excess wash was gently wicked off the coverslip, and then coverslips were squashed onto the mounting media. The chamber was sealed with nail polish and samples were imaged the same day.

Epifluorescence microscopy was used to obtain 4-color z-stacks of each spindle at ×100 magnification. The chromatin area was measured at the midplane of the spindle using Otsu's method to threshold to create a binary image, which was then eroded and dilated. The number of kinetochores was manually determined by counting the GFP puncta throughout each z-stack.

## Reporting summary

Further information on research design is available in the Nature Portfolio Reporting Summary linked to this article.

## Data availability

All other data used in this study are available upon request from the corresponding author. Source data are provided with this paper.

## Code availability

Image processing algorithms and numerical codes are described in the "Methods" and Supplementary Information file.

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

## Acknowledgements

We thank Jake DeLuca, Keith DeLuca, and Jeanne Mick for the CENPA-GFP HeLa cell line, Kayoko Hayashihara, and Kiichi Fukui for helpful discussions regarding chromosome isolation, Nachama Sterm for assisting with chromosome counting in purified fractions, Venecia Valdez for the Eg5 antibody, Gary Laevsky and the Confocal Imaging Facility, and all members of the Petry, Shaevitz, and Stone labs for advice and input. B.G. was supported by PD Soros fellowship, NSF GRFP DGE-2039656, and Wallace Memorial Honorific Fellowship. S.U.S. was supported by NIH NCI NRSA 1F31CA236160 and NHGRI training grant 5T32HG003284. M.R.K. was supported by NIGMS training grant T32GM007388. This work was funded by NIH 1DP2GM123493, NIH R01 GM141100, Pew Scholars Program 00027340, Packard Foundation 201440376, CPBF NSF PHY-1734030, and Wilke Family Foundation/O'Brien Family Fund for Health Research.

## Author contributions

B.G. did the theoretical and numerical analysis with assistance from H.A.S., designed and performed experiments, and analyzed experimental data. S.U.S. purified chromosomes, designed and performed experiments, and analyzed experimental data. M.R.K. and A.H. assisted in testing chromosome isolation protocols and preparing extract. S.U.S. and B.G. wrote the manuscript. H.A.S., J.W.S., and S.P. contributed to the research design, mentoring, and writing the manuscript.

## Competing interests

The authors declare no competing interests.
