## [Peer Review File · Nature Communications]

Reviewers' Comments:

Reviewer #1:

Remarks to the Author:

In this manuscript titled "Acentrosomal spindles assemble from branching microtubule nucleation near chromosomes" by Gouveia et al., the authors present data that elucidates the source of microtubules within the mitotic/meiotic spindle. This work represents an extension of previous efforts from the PI and her lab which established microtubule-mediated microtubule nucleation (or "branching" nucleation) as a phenomenon that occurs in metazoan systems. Though others have since confirmed this initial observation, it remains unclear as to whether branching microtubule nucleation actually plays a role in the assembly of the mitotic spindle, where the incredibly dense nature of the microtubule array has precluded definitive visualization and observation of individual branching events. In this work, the authors attempt to circumvent this problem by returning to the extract system and visualizing microtubule nucleation around isolated individual HeLa cell chromosomes (instead of the pseudo-tetraploid genome of whole *Xenopus laevis* nuclei typically used in such studies). This results in a relative reduction of microtubule density and, when combined with TIRF microscopy and its shallow excitation depth, further reduces the system's observable complexity to enable characterization of microtubule nucleation around the chromosome and its kinetochore with high spatiotemporal resolution. Using this approach, the authors go on to show that microtubule nucleation is a microtubule-seeded event that is initiated near chromosomes after a de novo generated microtubule collides with the kinetochore/chromosome. To determine whether the resulting arrays are indeed branched and generated by branching nucleation, the authors develop a model that describes with their results. They go on to show that biochemical perturbation of branching nucleation inhibits the formation of arrays around isolated chromosomes, adding credence to their claim that these arrays are indeed the result of branching nucleation. Lastly, they show that spindles can form around groups of isolated chromosomes when motor function is unperturbed. Unfortunately, though all of the results presented support the conclusion that microtubules are likely nucleated via a branching mechanism, in the absence of direct visualization of microtubule branching events, these data have to be more compelling and this threshold has not quite been met. However, my concerns would be assuaged pending the results of a few additional experiments as suggested below.

Major Questions/Comments/Concerns:

1) The results in Figure 1 are described in a presumptive way that undermines the rationale for the rest of the experiments performed, i.e. if you already know that the arrays are branched, why generate a model to prove it? I would instead suggest that the authors describe the results for this figure by omitting the word "branched" as there is no definitive proof that the arrays generated are indeed interconnected. Furthermore, I would urge caution in describing these results in the context of meiotic spindle assembly (as in lines 113-114). "These results indicate that microtubule nucleation is initiated near and towards kinetochores of isolated chromosomes" would be more accurate. Making these changes would also allow the authors to provide a more logically cogent transition to the next set of data which, at present, doesn't exist. As written, it is unclear why the authors are constructing a model at the point in the manuscript in which the model is introduced.

2) How does the proposed model differentiate between a branched network, presumably containing microtubules linked via nucleating complexes, versus a radial array of microtubules either unlinked, or tethered to the surface of the chromatin? If the authors' main argument is that it's the asymmetry in the directionality of the microtubule growth towards the chromatin/kinetochore, the data would be made more compelling if the authors provided evidence that motor activity, particularly that of the microtubule-clustering, pole-focusing motor cytoplasmic dynein, has indeed been inhibited by the addition of 0.5 mM of sodium orthovanadate (also, please provide a rationale or reference as to why this concentration was chosen). The authors should consider supplementing the vanadate with an established dynein-inhibitor, like p150-CC1 or demonstrating that motor activity is indeed inhibited using gliding motility assays, or by comparing the results of vanadate-only inhibition versus inhibition with a cocktail containing the dynein inhibitor p150-CC1 and a kinesin-5 inhibitor like STLC. If, however, the authors' main argument is something else, it should be better articulated in the text.

3) As presented in both main and supplemental figures, panels containing images of merged TIRF channels are difficult to interpret. For each figure, the authors need to include individual greyscale representations of each channel along with the merged image to show the data in a more unbiased and interpretable way.

4) The authors' claim that there is a RanGTP gradient formed around their isolated chromosomes. As this is central to their nucleation model, this should be experimentally validated using published probes (e.g. Kalab et al, 2002; Maresca et al, 2009). If these probes do indeed reveal a gradient, the authors should add Rant24N to block its formation and assess the effect on branched MT nucleation around the chromosome.

Minor Questions/Comments/Concerns:

1) Each channel in the supplemental movies should be adjusted for brightness and contrast. It's possible that it's an issue with my monitor, but I could not see any signal in the chromatin and EB1 channels in Movie S1.

2) For Figure 2, consider adding more detail to the legend to make it more easily interpreted and stand-alone.

3) In Figure 5, the authors need to better describe what constitutes a "pole" as it is not terribly clear in the included images. Addition of a pole marker, such as directed labeled anti-NuMA antibodies would enable a more quantitative assessment of these data.

Reviewer #2:

Remarks to the Author:

This manuscript by Gouveia et.al. addresses the question of how microtubules are nucleated in acentrosomal spindles, by reconstituting aspects of this process in *Xenopus* extracts. They purify chromosomes from HeLa cells, immobilize them on a slide, flow in *Xenopus* extract, and analyze microtubule nucleation. Using vanadate to block the action of molecular motors (and other ATPases), they are able to see robust branching microtubule nucleation occurring near chromosomes; they also developed a theoretical model that suggested that this process could be stimulated by releasing effectors near chromosomes, forming a concentration gradient that biases branching nucleation. Depletion of either TPX2 or augmin prevented nucleation around chromosomes in this assay, consistent with this hypothesis. Finally, the authors demonstrated that in the absence of vanadate, microtubules could reorganize into spindle-like arrays, recapitulating aspects of acentrosomal spindle assembly.

This paper reports numerous findings that will be of interest to the field. The discovery that branching microtubule nucleation can occur near chromosomes, and the resulting model (that the release of effectors could drive this process) represent a newly-described mechanism that may contribute to acentrosomal spindle assembly. However, I have a number of concerns that should be addressed before publication of this interesting study.

Major points:

1. The data in this paper is interesting, but the conclusions are vastly overstated in many places in the manuscript. Therefore, the authors need to be more open about the limitations of their study when discussing their data and revise their conclusions accordingly. Moreover, they need to avoid statements that imply that their findings are universally true of all acentrosomal spindles. This is a very artificial system that does not recapitulate all aspects of meiotic spindle assembly; while the findings may be relevant to the *Xenopus* system, it is not clear if all acentrosomal spindles form this way. Therefore, the authors need to revise bold claims that imply that their findings are universally true (some examples below). Moreover, the authors need to include a discussion of other models of acentrosomal spindle assembly, specifically acknowledging that there are systems where microtubule nucleation does not appear to originate at chromosomes. I think it is fine for the authors to speculate that branching microtubule nucleation COULD contribute to microtubule

generation in these other systems (as a secondary mechanism). However, implying that branching microtubule nucleation around chromosomes is the main source of microtubule generation in all acentrosomal spindles ignores a large body of work by others on in vivo models.

- The title "Acentrosomal spindles assemble from branching microtubule nucleation near chromosomes" implies that this is true of all acentrosomal spindles. However, nucleation in mouse oocytes initially occurs at non-centrosomal MTOCs (that are not initially found near chromosomes; Schuh and Ellenberg, 2007). In worm oocytes, the largest concentration of microtubules in the early stages of spindle assembly is near the disassembling nuclear envelope and not the chromosomes (Wolff et.al. 2016); since there is no disassembling NE in the Xenopus system, this aspect of meiotic progression is not recapitulated in this system. It is certainly possible that the microtubules generated away from chromosomes (in mouse/worms) could then come into contact with the chromosomes and stimulate branching microtubule nucleation, similar to what is observed in the current study. However, the suggestion that most microtubule nucleation is near chromosomes in acentrosomal spindles is an overstatement, given our current understanding of other systems.

- Lines 86-88: "branching microtubule nucleation....provides the main source of microtubules in acentrosomal spindles." Same comment as above - this may not be true of all acentrosomal models.

- Lines 113-114: "These results demonstrate that microtubule nucleation for meiotic spindle assembly is initiated near and towards kinetochores." It is unclear if this also occurs during meiosis in vivo, especially since meiotic chromosomes are not used in this study.

- Lines 207-209: "These results suggest that branching microtubule nucleation...is the key pathway to generate microtubules from chromosomes". Although the authors have evidence for this statement in extracts, this sentence is too broad. Similarly, the sentence on lines 209-211 should be changed from "meiotic Xenopus spindles" to "meiotic spindles formed in Xenopus extracts", since things may be different in Xenopus oocytes in vivo.

- Lines 273-275: "We show that microtubule generation around chromosomes during meiosis is attributable to RanGTP-mediated branching microtubule nucleation." This hasn't been shown in any other system, so is too strong as a blanket statement.

2. One limitation of this study that should be discussed in more detail is that the chromosomes are not the same as in vivo chromosomes. They are purified from HeLa cells (so are not meiotic) and it is also unclear if the purification procedure preserves all chromosomal proteins and proper chromosome compaction/organization. Also, the DNA is visualized with DAPI (which intercalates into DNA) and when Xenopus extract is flowed in, it is likely that Xenopus proteins also bind to these chromosomes; these experimental features mean that these chromosomes may not have entirely normal chromosome structure. This is something that should be discussed because the authors show that only ~30% of the chromosome clusters stimulate microtubule nucleation. Is this because the other clusters are not competent for nucleation? To understand the limitations of the assay, these issues should be discussed. Related to this point, the authors argue that microtubules can penetrate chromosome arms (Figure S4) but if the chromosomes are not condensed properly, this conclusion is called into question. The authors should therefore consider these issues related to chromosome structure/composition when interpreting their experiments.

3. Figure 1: The authors argue that microtubule branches initiate near kinetochores based on the observation that "50% of early nucleation events occurred within 4µm of a kinetochore" (lines 111-112). The way this section is written seems to imply that there is something special about kinetochores that might stimulate branching. However, is it possible that this is what you would expect by random chance, if microtubule branching is simply stimulated by chromatin? (i.e. Given the average spacing of kinetochores within a chromosome mass, could 50% of events be expected to be near a kinetochore just by chance?). If the authors really want to argue that nucleation is preferred near kinetochores, they need to provide experimental data to support this assertion. For example, an ideal experiment would be to do similar experiments on purified chromosomes lacking kinetochores, and then assess whether microtubule nucleation near these chromosomes is similar or different to the data presented in Figure 1. If such experiments are not experimentally feasible, the authors should adjust the claims presented in the manuscript.

4. The authors claim in many places that microtubule nucleation is initiated near chromosomes. However, multiple experiments in their paper show examples of microtubules first being nucleated

elsewhere. For example, Figure S1 shows that microtubules initially nucleate randomly but then nucleation can be amplified when a mother microtubule comes near a chromosome (rather than the chromosome itself driving the initial nucleation events). There is no evidence in the paper that all nucleation originates from chromosomes, which is implied in the text in a number of places. Thus, the authors should be more careful about their wording when describing these experiments - their data is consistent with chromosomes stimulating/amplifying nucleation, but is inconsistent with chromosomes driving initial nucleation. Some examples are:

- Lines 113-114: "These results demonstrate that microtubule nucleation for meiotic spindle assembly is initiated near and towards kinetochores." (Despite the authors noting earlier that, in this experiment, the initial nucleation was randomly throughout the imaging field - lines 103-105).
- Lines 215-216: "We observed that the initial microtubule nucleation events are still branches that grow near and towards kinetochores (Fig. 5a)". However, in the stills in Figure 5a, there are microtubules present near chromosomes in the timepoints before branches begin to form (- 1.5min). I would consider these to be the initial microtubule nucleation events, with the subsequent branching events amplifying this nucleation.
- Figure 6A: In the starting image, before microtubules are concentrated around the chromosomes, there are lots of microtubules already present in the extract.
- Movies S3 and S4 (control depletions): Many microtubules nucleate away from the chromosomes in the imaging field, and some even appear to form larger organized arrays, again showing that chromosomes are not driving all nucleation.

5. For the data shown in Figures 2A, 2B, 3A, and 3B, it is unclear if the experiments and measurements were done in the presence of vanadate. The images in Figures 2A and 3A look like the types of branched structures that form in the presence of vanadate (so I am assuming this is the case), but the authors do not specify whether this drug was used for the data shown in these figures (which was then compared to the theoretical models, as evidence that the models were correct). This is important because the process of microtubule generation may be very different in the presence of vanadate, where ATPases such as katanin and kinesin-13s are inhibited. Thus, if vanadate was used, it is unclear that the computational models represent appropriate representations of microtubule generation. Related to this, the graph presented in Figure 3B (showing the number of microtubules generated over time), is very different from the graph in Figure 6E. At 10 minutes, the graph in 3B shows the total number of microtubules plateauing in the range of 40-50 microtubules, while 6E shows it plateauing in the hundreds. These discrepancies add to the concern that the modeling in Figures 2 and 3 does not represent what is actually happening during microtubule nucleation in acentrosomal systems. A better discussion of these issues would be helpful for the reader to evaluate the claims of the manuscript.

6. The claim that there is an "emergent bipolar structure" in the assay shown in Figure 5 is not convincing based on the data presented. I understand that immobilization of the chromosomes to the slide affects the final spindle structure, but even so, the microtubule structures formed do not resemble normal acentrosomal spindles, so the claim that the authors are "reconstituting acentrosomal spindle assembly and organization" (line 212) is overstated. Moreover, this issue is compounded by the fact that I do not think that the quantification in Figure 5C is at all convincing, since it is based on counting the number of poles by eye, using images like the ones in Figure 5B. Since a pole marker is not used, it is impossible to tell how many organized poles are present in these highly disorganized structures (for example, the image labeled "bipolar" does not appear to me to have two organized poles, calling the whole quantification scheme into question; the structure in Movie S6 also does not look stably bipolar). Therefore, it is essential that the authors image a pole marker and use it to count the number of organized poles in their images, to better support their claims. Also related to this figure, the image in 5D (generated in an assay where the chromosomes were not immobilized) is slightly more convincing, as the structure has more distinct-looking poles. However, there is only one example of this result. If the authors want to use this data to support the assertion that they are reconstituting bipolar spindles, they need to image many more structures formed in 3-dimensional bulk extract and do a similar pole number quantification (using a pole marker).

Minor points:

- Abstract line 25: "depleting" should be "inhibiting" since vandate rather than immunodepletion

was used

- Lines 76-78: The authors state that previous modeling studies have focused only on how microtubule nucleation sustains the steady-state metaphase spindle. However, Loughlin et.al. (2010) modeled spindle assembly with different nucleation parameters (RanGTP vs. microtubule amplification, Figure 3). Prior studies should be discussed more accurately.
- Lines 219-220: The authors state that "the majority of spindles were bipolar (Fig. 5C)" but fewer than 50% were listed as bipolar on the relevant graph.
- It would be informative to inhibit Aurora B (the other microtubule nucleation pathway, as noted in lines 66-71) and assess the effects on microtubule nucleation in the presented assays. I suspect the authors have done this experiment since they provide information on Aurora inhibition in the Materials and Methods section (lines 542-547), so if they have these data I think they would be interesting to include.
- Figure 1: The legend does not state how kinetochores are labeled - this information should be added.
- Figure 2B and 3B: for the graphs on the right, the y-axis label should be "number of microtubules" to help the reader, and to be consistent with the graphs in Figure 6B and 6C.
- The authors might consider merging Figures 2 and 3. Although not an essential revision, this reviewer thought that it might be nice for the reader to see the models in the same figure, to more easily compare them.
- Figure 4B: The y-axis label should be more specific. I didn't know what "P" referred to until I looked at the legend. Change to "Fraction of chromosomes with networks" or something similar.
- Figure 5E: This model is confusing. It has astral microtubules and dots that look like centrosomes, which is not reflective of acentrosomal spindle organization.
- For Movies S3 and S4, the legends state that the controls are on the right, but they are on the left.

Reviewer #3:

Remarks to the Author:

My expertise is in modeling & computation so and I will not comment on the experimental methodologies.

Overview: The authors' consider a combined experimental & modeling study of microtubule nucleation. By comparing experiments and models, the authors' hypothesize that RanGTP-mediated branching microtubule nucleation in the vicinity of chromosomes provides the main source of microtubules in acentrosomal spindles.

Methodology: Overall I assess the modeling methodology to be appropriate but quite standard (see for example Stochastic Processes in Cell Biology, Bressloff). The use of the Laplace transform and its numerical inverse is well justified in obtaining solutions to the model. The qualitative agreement between the experiment and model appears to be good to the eye (see Fig 2 & 3). I didn't get a strong sense that there was interplay between the model and experimental work, where it was been used to guide or inform the experiments.

Significance: I will it to other reviewers to comment on the significance of the experiments. I thought this was a nice study with good qualitative agreement between the model and data. I did not feel that the authors' made a strong case for the importance of this work. It felt routine in that a model was quickly devised to provide agreement with some measurements.

Overall Quality: Well written and generally high quality manuscript. The supplemental information was detailed enough to reproduce work.

Reviewer #1 (Remarks to the Author):

In this manuscript titled “Acentrosomal spindles assemble from branching microtubule nucleation near chromosomes” by Gouveia et al., the authors present data that elucidates the source of microtubules within the mitotic/meiotic spindle. This work represents an extension of previous efforts from the PI and her lab which established microtubule-mediated microtubule nucleation (or “branching” nucleation) as a phenomenon that occurs in metazoan systems. Though others have since confirmed this initial observation, it remains unclear as to whether branching microtubule nucleation actually plays a role in the assembly of the mitotic spindle, where the incredibly dense nature of the microtubule array has precluded definitive visualization and observation of individual branching events. In this work, the authors attempt to circumvent this problem by returning to the extract system and visualizing microtubule nucleation around isolated individual Hela cell chromosomes (instead of the pseudo-tetraploid genome of whole *Xenopus laevis* nuclei typically used in such studies). This results in a relative reduction of microtubule density and, when combined with TIRF microscopy and its shallow excitation depth, further reduces the system’s observable complexity to enable characterization of microtubule nucleation around the chromosome and its kinetochore with high spatiotemporal resolution. Using this approach, the authors go on to show that microtubule nucleation is a microtubule-seeded event that is initiated near chromosomes after a de novo generated microtubule collides with the kinetochore/chromosome. To determine whether the resulting arrays are indeed branched and generated by branching nucleation, the authors develop a model that describes with their results. They go on to show that that biochemical perturbation of branching nucleation inhibits the formation of arrays around isolated chromosomes, adding credence to their claim that these arrays are indeed the result of branching nucleation. Lastly, they show that spindles can form around groups of isolated chromosomes when motor function is unperturbed. Unfortunately, though all of the results presented support the conclusion that microtubules are likely nucleated via a branching mechanism, in the absence of direct visualization of microtubule branching events, these data have to be more compelling and this threshold has not quite been met. However, my concerns would be assuaged pending the results of a few additional experiments as suggested below.

We thank the reviewer for their careful reading and detailed understanding of our manuscript. In our response that follows, the line numbers referenced refer to the manuscript without track changes.

We first wish to address the general point raised about the “absence of direct visualization of microtubule branching events”. We feel that such direct visual evidence is one of the strongest points of our paper. Therefore, in addition to Figs. 1b, 4a, 5a, and Movies S1, S5, we now include several additional time series images that clearly show new microtubules branching off preexisting microtubules near chromosomes, resulting in polar arrays (**Fig. S1**). This is definitively a microtubule-seeded nucleation process, which in the context of Fig. 1 of the paper we now call “microtubule-dependent nucleation”.

Major Questions/Comments/Concerns:

1) The results in Figure 1 are described in a presumptive way that undermines the rationale for the rest of the experiments performed, i.e. if you already know that the arrays are branched, why generate a model to prove it? I would instead suggest that the authors describe the results for this figure by omitting the word “branched” as there is no definitive proof that the arrays generated are indeed interconnected.

We agree with the reviewer’s point that the biochemical perturbations of known branching factors (Fig. 2) and model (Figs. 3-4) are rendered less important if we claim that the *specific* mechanism at play is branching nucleation at the outset. Therefore, we now use the more general language of “microtubule-dependent nucleation” around Fig. 1 and *hypothesize* that the specific mechanism of SAF-mediated microtubule branching nucleation is at play, which we proceed to confirm using biochemical perturbation experiments (Fig. 2) and modeling (Figs. 3-4) (**Fig. 1, Lines 109-122**).

Furthermore, we hope the additional images we now provide (**Fig. S1**) justify to the reviewer why we initially described the observed microtubule arrays as “branched”; that is, polar arrays in which new microtubules nucleate off preexisting ones. It is visually apparent by TIRFM data alone. However, simply describing an observation is not sufficient, so the biochemical perturbations and modeling are still necessary to complement and strengthen the direct TIRFM observations.

Furthermore, I would urge caution in describing these results in the context of meiotic spindle assembly (as in lines 113-114). “These results indicate that microtubule nucleation is initiated near and towards kinetochores of isolated chromosomes” would be more accurate.

We agree and have made the appropriate wording change (**Lines 118-119**).

Making these changes would also allow the authors to provide a more logically cogent transition to the next set of data which, at present, doesn’t exist. As written, it is unclear why the authors are constructing a model at the point in the manuscript in which the model is introduced.

We have now made these changes (**Lines 121-124, Lines 137-138**). The observation of microtubule-dependent nucleation near chromosomes in Fig. 1 now begets the *hypothesis* that the specific mechanism of SAF-mediated microtubule branching nucleation is the dominant contribution to the phenomena observed in Fig. 1, which we test by performing biochemical perturbation experiments (Fig. 2) and making a model (Figs. 3-4).

2) How does the proposed model differentiate between a branched network, presumably containing microtubules linked via nucleating complexes, versus a radial array of microtubules either unlinked, or tethered to the surface of the chromatin?

We appreciate this important question. In the absence of motors, we observe microtubule-dependent microtubule networks that grow towards chromosomes from a random *de novo*

microtubule (Fig. 1, Fig. S1). These networks are uniformly polar, *not* radial, due to the narrow branching angle at which a new microtubule nucleates off a preexisting one. Indeed, our model *assumes* that this observed polarity is maintained and that all microtubules formed (besides the *de novo* one) come from a nucleating complex on a preexisting microtubule, i.e., microtubules are always linked during their lifetime. Indeed, with motors inhibited, we observe essentially no microtubule transport in our TIRFM assay, and so the assumption that they remain linked is valid.

A radial network of unlinked microtubules or a network tethered to the surface of the chromatin would therefore be qualitatively inconsistent with our observations and model. The fact that we visually observe these polar networks and the fact that their plus-end organization is consistent with a model of branching microtubule nucleation in a SAF gradient (Fig. 4) is sufficient to rule out these other scenarios.

If the authors' main argument is that it's the asymmetry in the directionality of the microtubule growth towards the chromatin/kinetochore, the data would be made more compelling if the authors provided evidence that motor activity, particularly that of the microtubule-clustering, pole-focusing motor cytoplasmic dynein, has indeed been inhibited by the addition of 0.5 mM of sodium orthovanadate (also, please provide a rationale or reference as to why this concentration was chosen). The authors should consider supplementing the vanadate with an established dynein-inhibitor, like p150-CC1 or demonstrating that motor activity is indeed inhibited using gliding motility assays, or by comparing the results of vanadate-only inhibition versus inhibition with a cocktail containing the dynein inhibitor p150-CC1 and a kinesin-5 inhibitor like STLC. If, however, the authors' main argument is something else, it should be better articulated in the text.

To be clear, our main argument is that the distribution of branched microtubules is spatially biased by the SAF gradient released by chromosomes, which is supported by the data in Fig. 4, particularly the asymmetric plus-end distribution which matches our theoretical prediction. Intuitively, because the SAF gradient is peaked at the chromosome, microtubules are more likely to nucleate in its vicinity, and because branching microtubule nucleation is autocatalytic, this effect is self-reinforcing. This does not depend specifically on microtubule growth, but rather the spatial biasing of branching microtubule nucleation due to the SAF gradient. We have now provided a better argument in the text around Fig. 4.

Indeed, this conclusion relies on inhibiting motor activity to preserve the polarity and inhibit reorganization and/or gliding of the resultant branched networks so they can be compared to our model. We use 0.5 mM vanadate to achieve this, which has been the standard protocol in our lab since the original branching assay was devised (Petry et al., *Cell*, 2013).

We are confident that vanadate was successful in inhibiting motors for two reasons. One: there was no active microtubule motion in the vanadate experiments (Figs. 1-4, Movies S1-S4) compared to the obvious, dramatic microtubule motion and rearrangement in the experiments without vanadate (Fig. 5-6, Movies S5-S6). In particular, note the rapid motion of background

microtubules undergoing persistent motor-driven gliding on the coverslip in Movie S5 compared to the orderly polymerization of microtubules in Movie S1. Two: vanadate has already been compared against the dynein inhibitors p150-CC1 (Petry et al., *Cell*, 2013) and ciliobrevin (Gai et al., *Soft Matter*, 2021) in the context of studying branched microtubule networks in our lab, with the results showing that either is sufficient to inhibit motor transport.

3) As presented in both main and supplemental figures, panels containing images of merged TIRF channels are difficult to interpret. For each figure, the authors need to include individual greyscale representations of each channel along with the merged image to show the data in a more unbiased and interpretable way.

We thank the reviewer for this observation. We now show all split channels and merged images for all TIRFM images (**All Figs.**), which should make the data clearer for all readers.

4) The authors' claim that there is a RanGTP gradient formed around their isolated chromosomes. As this is central to their nucleation model, this should be experimentally validated using published probes (e.g. Kalab et al, 2002; Maresca et al, 2009). If these probes do indeed reveal a gradient, the authors should add RanT24N to block its formation and assess the effect on branched MT nucleation around the chromosome.

This is an excellent suggestion. In fact, in our lab we are currently pursuing the follow up project of systematically changing both the length scale and magnitude of the RanGTP gradient and visualizing it directly in our system to see if the resultant branched networks change according to our model. Of course, this requires a plethora of new experiments as well as using FRET-based probes, which falls outside the scope of this current study.

That being said, in order to address the reviewer's point, we sought instead to visualize the SAF gradient. Specifically, we chose the known SAF TPX2. The SAF gradient forms as a downstream consequence of the RanGTP gradient, and is actually more relevant to our model since it is what our model directly predicts for a given branched network organization. To achieve this, we immunodepleted endogenous TPX2 from extract and added back purified GFP-TPX2 at a physiological concentration of 100 nM. We then proceeded with our surface-based chromosome assay using this modified extract.

We clearly observe GFP-TPX2 enrichment near the chromosomes (**Fig. 4b left**). By plotting GFP-TPX2 intensity as a function of the distance from the chromosome using radial bins of 2 μm , we see the emergence of an exponential gradient. By averaging over 7 chromosomes across 2 different extract preparations, we found via linear regression that an exponentially decaying function with length scale $\lambda = 23 \pm 2 \mu\text{m}$ best fit our data (**Fig. 4b right**). This is reassuring because our previous estimate of the SAF gradient was $\lambda \approx 24 \mu\text{m}$, which we inferred as a fit parameter to our plus-end distributions.

We note that the length scale of the gradient is insensitive to the amount of microtubules nucleated in its vicinity (**Fig. S4**), which reassures us that the dominant feature we are

measuring is the concentration field of free, unbound TPX2. This is the case because TPX2 is initially bound by importins and is uniformly soluble in a sequestered state. Near chromosomes, RanGTP will release TPX2, and because it can self-associate (King et al., Nat Comms, 2020), it will tend to form bright puncta when freed, which is what we observe.

Thus, not only do we now have clear visual evidence of a SAF gradient in our system, we now have a direct measurement of the length scale of the gradient, reducing the number of fit parameters in our model to 1.

Minor Questions/Comments/Concerns:

1) Each channel in the supplemental movies should be adjusted for brightness and contrast. It's possible that it's an issue with my monitor, but I could not see any signal in the chromatin and EB1 channels in Movie S1.

We thank the reviewer for pointing this out. We have now adjusted contrast and brightness the movies further to make them more easily interpretable (*All Movies*).

2) For Figure 2, consider adding more detail to the legend to make it more easily interpreted and stand-alone.

We have now made the Fig. 3 (previously Fig. 2) legend more detailed and improved the detail of the text surrounding it (*Fig. 3, Lines 136-186*).

3) In Figure 5, the authors need to better describe what constitutes a "pole" as it is not terribly clear in the included images. Addition of a pole marker, such as directed labeled anti-NuMA antibodies would enable a more quantitative assessment of these data.

We agree with the reviewer that the poles generated in the TIRFM assay are not obviously apparent. Indeed, we suspect that this is due to the immobilization of the chromosomes on the coverslip. This is a substantial perturbation to the mechanics of spindle assembly, as the chromosomes cannot rearrange to accommodate the self-organizing forces imparted by motors. This likely leads to considerable stress and unstable poles. Another limitation is that we are only imaging in the TIRF field and are thus not observing the dynamics above the coverslip that might make the poles more visibly apparent. We have now discussed these points in the manuscript (*Lines 245-249*).

In order to provide more evidence that our system is capable of generating self-organized bipolar spindles, we performed new experiments using our bulk spindle assembly assay. We now present data taken from a sample of $n = 47$ chromosome clusters across 6 different extract preparations that formed multipolar spindles (*Fig. 5c-d, Fig. S5, Lines 249-257*). This confirms that our system is capable of reproducibly forming convincing bipolar spindles.

Unfortunately, we do not have a labeled pole marker (e.g. NuMA or an anti-NuMA antibody) readily available to add to our experiments, and making one and testing it would add complexity beyond the scope of our study's focus. Furthermore, there is precedent in the literature already for identifying poles using just fluorescently-labeled microtubules (Petry et al., PNAS, 2011; Schuh and Ellenberg, Cell, 2007; others). Now that we have made more bulk spindles with more visibly obvious poles, we believe this is enough to alleviate any doubts the reviewer may have about the ability of our system to reconstitute acentrosomal spindle organization.

Reviewer #2 (Remarks to the Author):

This manuscript by Gouveia et.al. addresses the question of how microtubules are nucleated in acentrosomal spindles, by reconstituting aspects of this process in *Xenopus* extracts. They purify chromosomes from HeLa cells, immobilize them on a slide, flow in *Xenopus* extract, and analyze microtubule nucleation. Using vanadate to block the action of molecular motors (and other ATPases), they are able to see robust branching microtubule nucleation occurring near chromosomes; they also developed a theoretical model that suggested that this process could be stimulated by releasing effectors near chromosomes, forming a concentration gradient that biases branching nucleation. Depletion of either TPX2 or augmin prevented nucleation around chromosomes in this assay, consistent with this hypothesis. Finally, the authors demonstrated that in the absence of vanadate, microtubules could reorganize into spindle-like arrays, recapitulating aspects of acentrosomal spindle assembly.

This paper reports numerous findings that will be of interest to the field. The discovery that branching microtubule nucleation can occur near chromosomes, and the resulting model (that the release of effectors could drive this process) represent a newly-described mechanism that may contribute to acentrosomal spindle assembly. However, I have a number of concerns that should be addressed before publication of this interesting study.

We thank the reviewer for their careful reading and impressively detailed understanding of our work. We address all the concerns raised in the following response. In our response that follows, the line numbers referenced refer to the manuscript without track changes.

Major points:

1. The data in this paper is interesting, but the conclusions are vastly overstated in many places in the manuscript. Therefore, the authors need to be more open about the limitations of their study when discussing their data and revise their conclusions accordingly. Moreover, they need to avoid statements that imply that their findings are universally true of all acentrosomal spindles. This is a very artificial system that does not recapitulate all aspects of meiotic spindle assembly; while the findings may be relevant to the *Xenopus* system, it is not clear if all acentrosomal spindles form this way. Therefore, the authors need to revise bold claims that imply that their findings are universally true (some examples below). Moreover, the authors need to include a discussion of other models of acentrosomal spindle assembly, specifically acknowledging that there are systems where microtubule nucleation does not appear to originate at chromosomes. I think it is fine for the authors to speculate that branching microtubule nucleation COULD contribute to microtubule generation in these other systems (as a secondary mechanism). However, implying that branching microtubule nucleation around chromosomes is the main source of microtubule generation in all acentrosomal spindles ignores a large body of work by others on in vivo models.

This is an important point, and we thank the reviewer for raising it. We agree that we were too brief with our language and scope of discussion. We have now improved our text throughout the manuscript to accommodate the suggestions the reviewer makes below.

- The title “Acentrosomal spindles assemble from branching microtubule nucleation near chromosomes” implies that this is true of all acentrosomal spindles. However, nucleation in mouse oocytes initially occurs at non-centrosomal MTOCs (that are not initially found near chromosomes; Schuh and Ellenberg, 2007). In worm oocytes, the largest concentration of microtubules in the early stages of spindle assembly is near the disassembling nuclear envelope and not the chromosomes (Wolff et.al. 2016); since there is no disassembling NE in the *Xenopus* system, this aspect of meiotic progression is not recapitulated in this system. It is certainly possible that the microtubules generated away from chromosomes (in mouse/worms) could then come into contact with the chromosomes and stimulate branching microtubule nucleation, similar to what is observed in the current study. However, the suggestion that most microtubule nucleation is near chromosomes in acentrosomal spindles is an overstatement, given our current understanding of other systems.

We have now changed the title of our paper to restrict its direct applicability to only *Xenopus* systems (**Title**).

Furthermore, we have expanded our paragraph discussing alternate mechanisms of acentrosomal spindle assembly in different systems. We note that in the Wolff et al., 2016 reference, the authors only observe early spindle assembly near the disassembling nuclear envelope in meiosis I. In meiosis II, which our extract is arrested in, they observe microtubule generation near chromosomes, consistent with our model (**Lines 301-310**).

- Lines 86-88: “branching microtubule nucleation...provides the main source of microtubules in acentrosomal spindles.” Same comment as above - this may not be true of all acentrosomal models.

We have now restricted that statement to specify *Xenopus* egg extract spindles only (**Lines 89-91**).

- Lines 113-114: “These results demonstrate that microtubule nucleation for meiotic spindle assembly is initiated near and towards kinetochores.” It is unclear if this also occurs during meiosis *in vivo*, especially since meiotic chromosomes are not used in this study.

We have now qualified that statement to specify that it is true for isolated mitotic chromosomes (**Lines 118-119**).

- Lines 207-209: “These results suggest that branching microtubule nucleation...is the key pathway to generate microtubules from chromosomes”. Although the authors have evidence for this statement in extracts, this sentence is too broad. Similarly, the sentence on lines 209-211

should be changed from “meiotic *Xenopus* spindles” to “meiotic spindles formed in *Xenopus* extracts”, since things may be different in *Xenopus* oocytes in vivo.

We have now restricted both those statements to highlight their relevance in *Xenopus* egg extracts only (**Lines 131-133**).

- Lines 273-275: “We show that microtubule generation around chromosomes during meiosis is attributable to RanGTP-mediated branching microtubule nucleation.” This hasn’t been shown in any other system, so is too strong as a blanket statement.

We have now removed that statement and restricted the wording around it to being true in *Xenopus* egg extract only (**Lines 311-318**).

2. One limitation of this study that should be discussed in more detail is that the chromosomes are not the same as in vivo chromosomes. They are purified from HeLa cells (so are not meiotic) and it is also unclear if the purification procedure preserves all chromosomal proteins and proper chromosome compaction/organization. Also, the DNA is visualized with DAPI (which intercalates into DNA) and when *Xenopus* extract is flowed in, it is likely that *Xenopus* proteins also bind to these chromosomes; these experimental features mean that these chromosomes may not have entirely normal chromosome structure. This is something that should be discussed because the authors show that only ~30% of the chromosome clusters stimulate microtubule nucleation. Is this because the other clusters are not competent for nucleation? To understand the limitations of the assay, these issues should be discussed.

Related to this point, the authors argue that microtubules can penetrate chromosome arms (Figure S4) but if the chromosomes are not condensed properly, this conclusion is called into question. The authors should therefore consider these issues related to chromosome structure/composition when interpreting their experiments.

These are excellent points that we now spend time addressing in our manuscript (**Lines 248-249, 351-356**).

3. Figure 1: The authors argue that microtubule branches initiate near kinetochores based on the observation that “50% of early nucleation events occurred within 4 μ m of a kinetochore” (lines 111-112). The way this section is written seems to imply that there is something special about kinetochores that might stimulate branching. However, is it possible that this is what you would expect by random chance, if microtubule branching is simply stimulated by chromatin? (i.e. Given the average spacing of kinetochores within a chromosome mass, could 50% of events be expected to be near a kinetochore just by chance?). If the authors really want to argue that nucleation is preferred near kinetochores, they need to provide experimental data to support this assertion. For example, an ideal experiment would be to do similar experiments on purified chromosomes lacking kinetochores, and then assess whether microtubule nucleation near these chromosomes is similar or different to the data

presented in Figure 1. If such experiments are not experimentally feasible, the authors should adjust the claims presented in the manuscript.

The reviewer is correct – we do not have any experimental evidence to support a claim that the kinetochores themselves specifically stimulate branching nucleation. In the text around Fig. 1, we were just making observations in the absence of any model using the labeled kinetochores as a convenient and important marker for the center of the chromosome. Indeed, our model relies only on chromatin-bound RCC1's ability to produce RanGTP, which releases SAFs in a gradient around chromosomes. We have now clarified this throughout our manuscript (**Lines 24, 87, 298, 316, 321, 325-327**).

However, that is not to say that the kinetochore is not important. There is evidence that kinetochores themselves can recruit γ TuRC directly via the Nup107-160 nucleoporin complex, allowing for microtubule nucleation directly off kinetochores (Mishra et al., NCB, 2010). It might be that this pathway is at play in our system, but given our results in Fig. 4, it would likely play only a supporting role. Indeed, the way to test this pathway would be to perform two similar experiments: one with chromosomes without kinetochores and another with just kinetochores. We have added this point to our discussion section (**Lines 328-333**).

Perhaps more importantly is that once a microtubule contacts a kinetochore with its plus-end, it is stabilized (Fig. 1b, Movie S1). This stabilized microtubule would then serve as an ideal template to generate branched microtubules near chromosomes via our proposed mechanism. The branches would automatically be directed towards kinetochores due to the shallow branching angle, which is conducive to kinetochore fiber formation. This is exactly what was found in human mitotic cells (David et al., JCB, 2019), where the authors showed that microtubules nucleated towards kinetochores in an augmin-dependent manner, which was found to be essential for kinetochore fiber maintenance in metaphase. We now discuss this in our discussion section as well (**Lines 322-325**).

4. The authors claim in many places that microtubule nucleation is initiated near chromosomes. However, multiple experiments in their paper show examples of microtubules first being nucleated elsewhere. For example, Figure S1 shows that microtubules initially nucleate randomly but then nucleation can be amplified when a mother microtubule comes near a chromosome (rather than the chromosome itself driving the initial nucleation events). There is no evidence in the paper that all nucleation originates from chromosomes, which is implied in the text in a number of places. Thus, the authors should be more careful about their wording when describing these experiments - their data is consistent with chromosomes stimulating/amplifying nucleation, but is inconsistent with chromosomes driving initial nucleation.

We thank the reviewer for this careful observation. Indeed, as we point out in Lines 106-111 and as the reviewer correctly notes, the initial microtubules that seed the formation of chromosomal branched networks nucleate randomly throughout the imaging field, independent of chromosomes. We refer to these microtubules as being nucleated "*de novo*". We agree that the

language “initiated at chromosomes” might be confusing in this context, so we have used more precise language throughout the text as detailed below.

Some examples are:

- Lines 113-114: “These results demonstrate that microtubule nucleation for meiotic spindle assembly is initiated near and towards kinetochores.” (Despite the authors noting earlier that, in this experiment, the initial nucleation was randomly throughout the imaging field - lines 103-105).

This has now been fixed in (**Lines 118-119**).

- Lines 215-216: “We observed that the initial microtubule nucleation events are still branches that grow near and towards kinetochores (Fig. 5a)”. However, in the stills in Figure 5a, there are microtubules present near chromosomes in the timepoints before branches begin to form (- 1.5min). I would consider these to be the initial microtubule nucleation events, with the subsequent branching events amplifying this nucleation.

This has now been fixed in (**Lines 237-241**).

- Figure 6A: In the starting image, before microtubules are concentrated around the chromosomes, there are lots of microtubules already present in the extract.

Indeed, background microtubule nucleation in *Xenopus* meiotic extract is highly variable. Some extract preps have a high background whereas others show essentially zero background activity. What matters for our study is that the microtubule networks generated near chromosomes have a density that is far above the background microtubule density, which is clearly the case in Fig. 6.

- Movies S3 and S4 (control depletions): Many microtubules nucleate away from the chromosomes in the imaging field, and some even appear to form larger organized arrays, again showing that chromosomes are not driving all nucleation.

Given the number of chromosomes in those fields of views and a length scale of $23 \pm 2 \mu\text{m}$ for the SAF gradient, it is not surprising that some networks will form farther away or in between chromosomes. However, it is still clear in Movies S3 and S4 that the largest chromosome clusters form the densest networks, consistent with our model (Fig. 6e).

5. For the data shown in Figures 2A, 2B, 3A, and 3B, it is unclear if the experiments and measurements were done in the presence of vanadate. The images in Figures 2A and 3A look like the types of branched structures that form in the presence of vanadate (so I am assuming this is the case), but the authors do not specify whether this drug was used for the data shown in these figures (which was then compared to the theoretical models, as evidence that the models were correct). This is important because the process of microtubule generation may be very different in the presence of vanadate, where ATPases such as katanin and kinesin-13s are

inhibited. Thus, if vanadate was used, it is unclear that the computational models represent appropriate representations of microtubule generation.

Related to this, the graph presented in Figure 3B (showing the number of microtubules generated over time), is very different from the graph in Figure 6E. At 10 minutes, the graph in 3B shows the total number of microtubules plateauing in the range of 40-50 microtubules, while 6E shows it plateauing in the hundreds. These discrepancies add to the concern that the modeling in Figures 2 and 3 does not represent what is actually happening during microtubule nucleation in acentrosomal systems. A better discussion of these issues would be helpful for the reader to evaluate the claims of the manuscript.

We thank the reviewer for bringing up this important point. Indeed, Figs. 1-4 display data from experiments done in the presence of vanadate to inhibit motors. This has now been made more explicit in the Figs. 1-4 legends.

The point of using vanadate was to eliminate the complexity of motor organization so as to focus on the dominant microtubule nucleation physics that underlie spindle assembly in this system. That way, we could compare the experimental results with our theoretical model that only takes microtubule polymerization and nucleation processes into account while neglecting motor transport. We are confident that vanadate was successful in inhibiting motors for two reasons. One: there was no active microtubule motion in the vanadate experiments (Figs. 1-4, Movies S1-S4) compared to the obvious, dramatic microtubule motion and rearrangement in the experiments without vanadate (Fig. 5-6, Movies S5-S6). In particular, note the rapid motion of background microtubules undergoing persistent motor-driven gliding on the coverslip in Movie S5 compared to the orderly polymerization of microtubules in Movie S1. Two: vanadate has already been compared against the dynein inhibitors p150-CC1 (Petry et al., *Cell*, 2013) and ciliobrevin (Gai et al., *Soft Matter*, 2021) in the context of studying branched microtubule networks in our lab, with the results showing that either is sufficient to inhibit motor transport. Therefore, we are confident that our theoretical model is directly applicable to the experiments when vanadate is present.

At the same time, being a nonspecific inhibitor of ATPases, vanadate may inevitably inhibit polymerases or severases that utilize ATP to actively tune microtubule polymerization. Therefore, the total microtubule mass will be affected, which will in turn quantitatively affect the number of microtubules generated since branching nucleation depends strongly on the amount of microtubule lattice present. We also speculate that motors, when not inhibited, pull out branched microtubules from their branch site, and thus enable a new round of branching microtubule nucleation to take place from the same site. These points may help explain the differences between Fig. 4c (previously Fig. 3b) and Fig. 6c. We are encouraged by the fact that, while Fig. 3c and Fig. 6c are quantitatively different, they are qualitatively similar in that they both show exponential amplification of microtubules within the first ~10 minutes followed by a plateau, which suggests similar nucleation physics are at play that are insensitive to ATP-related activity. We have now added these discussion points to our manuscript (**Lines 357-365**).

6. The claim that there is an “emergent bipolar structure” in the assay shown in Figure 5 is not convincing based on the data presented. I understand that immobilization of the chromosomes to the slide affects the final spindle structure, but even so, the microtubule structures formed do not resemble normal acentrosomal spindles, so the claim that the authors are “reconstituting acentrosomal spindle assembly and organization” (line 212) is overstated. Moreover, this issue is compounded by the fact that I do not think that the quantification in Figure 5C is at all convincing, since it is based on counting the number of poles by eye, using images like the ones in Figure 5B. Since a pole marker is not used, it is impossible to tell how many organized poles are present in these highly disorganized structures (for example, the image labeled “bipolar” does not appear to me to have two organized poles, calling the whole quantification scheme into question; the structure in Movie S6 also does not look stably bipolar). Therefore, it is essential that the authors image a pole marker and use it to count the number of organized poles in their images, to better support their claims.

Also related to this figure, the image in 5D (generated in an assay where the chromosomes were not immobilized) is slightly more convincing, as the structure has more distinct-looking poles. However, there is only one example of this result. If the authors want to use this data to support the assertion that they are reconstituting bipolar spindles, they need to image many more structures formed in 3-dimensional bulk extract and do a similar pole number quantification (using a pole marker).

We agree with the reviewer that the poles generated in the TIRFM assay are not obviously apparent. Indeed, we suspect that this is due to the immobilization of the chromosomes on the coverslip. This is a substantial perturbation to the mechanics of spindle assembly, as the chromosomes cannot rearrange to accommodate the self-organizing forces imparted by motors. This likely leads to considerable stress and unstable poles. Another limitation is that we are only imaging in the TIRF field and are thus not observing the dynamics above the coverslip that might make the poles more visibly apparent. We have now discussed these points in the manuscript (**Lines 245-249**).

In order to provide more evidence that our system is capable of generating self-organized bipolar spindles, we performed new experiments using our bulk spindle assembly assay. We now present data taken from a sample of $n = 47$ chromosome clusters across 6 different extract preparations that formed multipolar spindles (**Fig. 5c-d, Fig. S5, Lines 249-257**). This confirms that our system is capable of reproducibly forming convincing bipolar spindles.

Unfortunately, we do not have a labeled pole marker (e.g. NuMA or an anti-NuMA antibody) readily available to add to our experiments, and making one and testing it would add complexity beyond the scope of our study’s focus. Furthermore, there is precedent in the literature already for identifying poles using just fluorescently-labeled microtubules (Petry et al., PNAS, 2011; Schuh and Ellenberg, Cell, 2007; others). Now that we have made more bulk spindles with more visibly obvious poles, we believe this is enough to alleviate any doubts the reviewer may have about the ability of our system to reconstitute acentrosomal spindle organization.

Minor points:

- Abstract line 25: “depleting” should be “inhibiting” since vandate rather than immunodepletion was used

We have now made that change (**Line 25**).

- Lines 76-78: The authors state that previous modeling studies have focused only on how microtubule nucleation sustains the steady-state metaphase spindle. However, Loughlin et.al. (2010) modeled spindle assembly with different nucleation parameters (RanGTP vs. microtubule amplification, Figure 3). Prior studies should be discussed more accurately.

We thank the reviewer for pointing this out. Indeed, Loughlin et al. do study different nucleation pathways in a detailed manner, however their focus was how those pathways affect properties of the resulting metaphase spindle at *steady-state*, such as spindle length and microtubule density profile. The initial condition that they use to achieve a metaphase spindle consists of two anti-parallel arrays of pre-nucleated microtubules (their Fig. 1a). This is an artificial initial condition that was chosen to speed up simulation time, which is why their results can only be strictly interpreted at the metaphase steady-state. We have clarified this further in the text (**Lines 80-82**).

- Lines 219-220: The authors state that “the majority of spindles were bipolar (Fig. 5C)” but fewer than 50% were listed as bipolar on the relevant graph.

This section has now been entirely reworded (**Lines 254-256**).

- It would be informative to inhibit Aurora B (the other microtubule nucleation pathway, as noted in lines 66-71) and assess the effects on microtubule nucleation in the presented assays. I suspect the authors have done this experiment since they provide information on Aurora inhibition in the Materials and Methods section (lines 542-547), so if they have these data I think they would be interesting to include.

We appreciate the reviewer’s eye for detail! We decided to remove that data, and have now removed the corresponding part from the methods section, since it was too preliminary. There is currently an ongoing follow-up project in our lab looking into the role of the CPC in chromosome-mediated microtubule nucleation. However, we decided that this constitutes a separate piece of work that warrants its own manuscript and requires the typical few years of further study until completion.

- Figure 1: The legend does not state how kinetochores are labeled - this information should be added.

We have added the labeling description to the text (**Line 98**) and Fig. 1 legend.

- Figure 2B and 3B: for the graphs on the right, the y-axis label should be “number of microtubules” to help the reader, and to be consistent with the graphs in Figure 6B and 6C.

We have made this change (*Figs. 3c and 4c*).

- The authors might consider merging Figures 2 and 3. Although not an essential revision, this reviewer thought that it might be nice for the reader to see the models in the same figure, to more easily compare them.

We appreciate the suggestion. We played with many different figure outlines and overlays and concluded that a figure combining both models would be informatively overwhelming.

- Figure 4B: The y-axis label should be more specific. I didn't know what “P” referred to until I looked at the legend. Change to “Fraction of chromosomes with networks” or something similar.

We have made this change (*Fig. 2b*).

- Figure 5E: This model is confusing. It has astral microtubules and dots that look like centrosomes, which is not reflective of acentrosomal spindle organization.

Thank you for pointing this out. We have now made a new and improved model schematic for Fig. 5e.

- For Movies S3 and S4, the legends state that the controls are on the right, but they are on the left.

We have made this change (*Movies S3-S4 legends*).

Reviewer #3 (Remarks to the Author):

My expertise is in modeling & computation so and I will not comment on the experimental methodologies.

Overview: The authors' consider a combined experimental & modeling study of microtubule nucleation. By comparing experiments and models, the authors' hypothesize that RanGTP-mediated branching microtubule nucleation in the vicinity of chromosomes provides the main source of microtubules in acentrosomal spindles.

We thank the reviewer for their careful reading and understanding of the quantitative aspects of our manuscript.

Methodology: Overall I assess the modeling methodology to be appropriate but quite standard (see for example Stochastic Processes in Cell Biology, Bressloff). The use of the Laplace transform and its numerical inverse is well justified in obtaining solutions to the model. The qualitative agreement between the experiment and model appears to be good to the eye (see Fig 2 & 3). I didn't get a strong sense that there was interplay between the model and experimental work, where it was been used to guide or inform the experiments.

We have now completely reworked the text so that it is more apparent how theory and experiment interplay with each other. Specifically, the observation of microtubule-dependent nucleation near chromosomes in Fig. 1 now begets the *hypothesis* that the specific mechanism of RanGTP-mediated microtubule branching nucleation is the dominant contribution to the phenomena observed in Fig. 1, which we test in part by using our model (Figs. 3-4).

Significance: I will it to other reviewers to comment on the significance of the experiments. I thought this was a nice study with good qualitative agreement between the model and data. I did not feel that the authors' made a strong case for the importance of this work. It felt routine in that a model was quickly devised to provide agreement with some measurements.

In addition to strengthening the interplay between model and data throughout the manuscript, we have now completely reworked the discussion section to better put our results in the context of the broader cytoskeleton field. We hope the novelty and importance of our work is more apparent in the revised manuscript.

Overall Quality: Well written and generally high quality manuscript. The supplemental information was detailed enough to reproduce work.

We thank the reviewer for their kind remarks.

Reviewers' Comments:

Reviewer #1:

Remarks to the Author:

This revised manuscript from Gouveia et al represents an improvement over the original submission as it now includes new data in which the spatial distribution of the RanGTP has been partially characterized and a better, more interpretable presentation of imaging data. However, the authors failed to label spindle poles as requested and the data still does not adequately support the original main conclusion of the work, that autocatalytic nucleation around chromosomes produces bona fide spindles. As such, the work simply recapitulates much of what is already known, that microtubule nucleation occurs around kinetochores and chromosomes in a TPX2-dependent manner (e.g. Tulu et al., 2006) and that the growth of microtubules around isolated chromosomes in extract is mostly directed toward the chromosome (e.g. Maresca et al., 2005). The novel observations of this paper are that augmin is also required and that the formation of the assemblies is consistent with an autocatalytic microtubule nucleation model. In the absence of definitive evidence suggesting that this mechanism is also responsible for generating bona fide bipolar spindles, these findings do not represent a significant enough advance to merit publication in Nature Communications. However, my concerns can be addressed by additional experiments and analyses.

The fact that the authors observe what appear to be bona fide and robust spindles forming around unanchored chromosomes in bulk extract (Fig. 5c) is not at all unexpected. However, bona fide spindles assembled around individual pairs of sister chromatids would be. It is quite likely that multiple chromosomes are required to generate robust spindles and that nascent microtubule assemblies generated around free floating chromosomes interact and coalesce in bulk extract, just like physically juxtaposed spindles and asters (e.g. Gatlin et al., 2009). In fact, the chromosome signal seems more robust in the images of fixed spindles assembled in bulk extract (compare chromosome channels in 5b & c), consistent with this possibility. To assuage my concerns, the authors need to count the number of chromatids in each spindle. If indeed a single chromatid (or pair of chromatids) can generate a "spindle" then this would be a more impactful finding and it would lend more credence to their central conclusion.

This leads to the question of "What is a spindle?", one that I ask because the data suggesting that "spindles" can form around isolated chromosomes is still underwhelming and the conclusions in this regard still overstated in this revised draft. In my view, two or three interacting aster-like microtubule assemblies around a chromatin blob doesn't quite meet the definition. Minimally, the authors need to correlate bipolarity with chromosome number for both the anchored chromatin case and in bulk extract spindles. They also need to provide additional characterization that the structures observed are indeed real bipolar spindles, starting with adding a pole marker to the mix to facilitate accurate counting of poles (i.e. labeled anti-NuMA antibodies), and then perhaps by showing some additional evidence of spindle-like properties, such as Eg5-dependent microtubule flux (or minimally midzone localization of Eg5 or some other marker of antiparallel microtubule overlap).

Reviewer #2:

Remarks to the Author:

The authors have done a thorough job responding to my comments and the revised manuscript is greatly improved. I think that this work makes an important contribution to the field. After my readthrough of the revised manuscript, I have a few final comments aimed at increasing the clarity of the manuscript (mostly edits to the figures to improve the data presentation). With these changes, I am supportive of publication of this interesting work.

Specific points:

- The authors did a good job altering the text to better emphasize/explain that de novo microtubule nucleation occurs, to clarify that not all microtubules in this system are templated off other microtubules. However, there was one remaining sentence that I still thought was confusingly worded (lines 116-117). Instead of stating that "These results demonstrate that

microtubule nucleation in this system is microtubule-dependent”, it would be more precise to change it to “most microtubule nucleation in this system” or something similar (since microtubules can be nucleated de novo, not all nucleation is microtubule-dependent).

- Reviewer 1 requested that all figures include individual greyscale representations for each channel – I strongly agree with this suggestion. The authors did alter their figures in response to Reviewer 1's comment, but the single color images they included are not greyscale (they are red on black, green on black, etc). These colored panels are much harder to interpret than greyscale images (the contrast is less obvious so it is hard to see the signal). I strongly encourage the authors to change all single-color images to greyscale (in every figure), and only use colors in the merges.

- In figure 1B, 5A, and S1, the numbers in the bottom row (that denote unique microtubule plus ends) are very small and hard to see - make them bigger if you want to include them.

- The text is very small in some of the figures – I suggest going through and increasing the font size where possible. Some examples include the labels on the axes of graphs, the text in Figure 4A, the labels in Figure 5A, the text in figure 6B-E, and the labels in Figure S1. The size of the entire Figure S3 could also be increased.

- The graph in Figure S3 has four traces but only three conditions noted on the graph legend. I am assuming based on the colors that two of these are control traces, but the figure legend does not explain why two controls are shown. Please clarify this somewhere (and also clarify why the two control traces are different).

- I found the layout of Figure S6A to be confusing. The legend states that the bottom right panels show merged images, but many of the other images appear to be merges as well. I think that it would be helpful to alter the organization and labels on this figure, to make it more clear what is being shown in each image (and edit the figure legend to more clearly explain this as well).

- In some of the sections that were edited, new references were added, but these were not added into the numbered reference list (they are just listed as “Author, et.al.” and not given a number or included in the reference list).

Reviewer #3:

Remarks to the Author:

The authors' response addressed all of my points.

Reviewer #1 (Remarks to the Author):

This revised manuscript from Gouveia et al represents an improvement over the original submission as it now includes new data in which the spatial distribution of the RanGTP has been partially characterized and a better, more interpretable presentation of imaging data. However, the authors failed to label spindle poles as requested and the data still does not adequately support the original main conclusion of the work, that autocatalytic nucleation around chromosomes produces bona fide spindles. As such, the work simply recapitulates much of what is already known, that microtubule nucleation occurs around kinetochores and chromosomes in a TPX2-dependent manner (e.g. Tulu et al., 2006) and that the growth of microtubules around isolated chromosomes in extract is mostly directed toward the chromosome (e.g. Maresca et al., 2005). The novel observations of this paper are that augmin is also required and that the formation of the assemblies is consistent with an autocatalytic microtubule nucleation model. In the absence of definitive evidence suggesting that this mechanism is also responsible for generating bona fide bipolar spindles, these findings do not represent a significant enough advance to merit publication in Nature Communications. However, my concerns can be addressed by additional experiments and analyses.

We thank the reviewer for their careful reading and detailed understanding of our work. Their comments have prompted us to conduct new experiments and analysis which we outline in detail below. We believe this newly revised manuscript sufficiently address all the raised concerns. All line numbers and figure numbers refer to the newly revised manuscript.

To summarize, we now include the requested immunofluorescent staining of NuMA on bulk spindles, enabling us to label poles unambiguously. We also immunostained spindles for Eg5. Additionally, by gathering more data on bulk spindles we established how emergent spindle polarity depends on both chromosome number and chromatin area, a novel result for the field.

Using our TIRF assay, we had already directly visualized autocatalytic branching events near purified chromosomes both in the absence of motors (Figs. 1, 4, S1) and in the presence of motors (Fig. 5) and showed that this required the *Xenopus* branching factors augmin and TPX2 (Fig. 2). Now, our new data (Figs. 6, S5) show that the spindles generated from this pathway in bulk extract are proper bona fide spindles that are biologically functional.

It is of course another challenge to directly confirm that branching nucleation is happening live in our bulk extract spindles. This is due to the limited resolution of bulk optical methods as well as the spindle getting dense with microtubules quickly, which was our rationale for doing the TIRF assays. Indeed, it is an ongoing challenge for the field to establish the molecular-scale regulation of microtubule nucleation pathways at the onset of acentrosomal spindle assembly. Further *ex vivo* studies akin to ours as well as *in vivo* work using super-resolution methods will be needed.

In conclusion, our current data now justifies the mechanism whereby *Xenopus* egg extract spindles assemble around chromosomes through branching nucleation and are subsequently organized by motors.

The fact that the authors observe what appear to be bona fide and robust spindles forming around unanchored chromosomes in bulk extract (Fig. 5c) is not at all unexpected. However, bona fide spindles assembled around individual pairs of sister chromatids would be. It is quite likely that multiple chromosomes are required to generate robust spindles and that nascent microtubule assemblies generated around free floating chromosomes interact and coalesce in bulk extract, just like physically juxtaposed spindles and asters (e.g. Gatlin et al., 2009). In fact, the chromosome signal seems more robust in the images of fixed spindles assembled in bulk extract (compare chromosome channels in 5b & c), consistent with this possibility. To assuage my concerns, the authors need to count the number of chromatids in each spindle. If indeed a single chromatid (or pair of chromatids) can generate a “spindle” then this would be a more impactful finding and it would lend more credence to their central conclusion.

We thank the reviewer for this insightful comment. To address this, we have now generated a larger data set of $n = 173$ z-stacks of bulk spindles with labeled kinetochores, allowing us to count the total number of chromosomes (i.e., half the number of chromatids) in each spindle. We find that an increasing number of chromosomes leads to an increasing number of spindle poles (**Fig. 6e**), a novel conclusion that was only achievable thanks to our reconstituted system, where we can generate a large size distribution of purified chromosome clusters all with labeled kinetochores.

We did not observe any isolated chromatids in any of our data. As suspected by the reviewer, we found that individual chromosomes could not generate bona fide bipolar spindles. Single chromosomes could only generate relatively small monopolar spindles, as shown below in an example.

Figure R1: Monopolar spindle generated by a single chromosome. White arrow labels the kinetochore pair. Scale bar is 10 μm .

The minimum number of chromosomes that we found capable of generating a bipolar spindle was 5. Therefore, we conclude from our data that there is indeed a minimum

number of chromosomes (greater than 1) required to form a bipolar spindle. This finding is not in conflict with our central conclusion that spindles form around chromosomes (plural) due to branching nucleation and subsequent self-organization by motors, but it does point to an interesting threshold, likely in microtubule mass, that must be met for multiple poles to be sustained.

We note that the chromosome signal being more robust in epifluorescence images (Figs. 6a-b) compared with TIRF images (Figs. 5a-c) is simply because chromosome sizes are on the micron scale, whereas in TIRF we are only resolving a region on the order of ~100 nm. Thus, most of the chromosome signal is not resolved in TIRF, leading to noisy images.

This leads to the question of “What is a spindle?”, one that I ask because the data suggesting that “spindles” can form around isolated chromosomes is still underwhelming and the conclusions in this regard still overstated in this revised draft. In my view, two or three interacting aster-like microtubule assemblies around a chromatin blob doesn't quite meet the definition. Minimally, the authors need to correlate bipolarity with chromosome number for both the anchored chromatin case and in bulk extract spindles. They also need to provide additional characterization that the structures observed are indeed real bipolar spindles, starting with adding a pole marker to the mix to facilitate accurate counting of poles (i.e. labeled anti-NuMA antibodies), and then perhaps by showing some additional evidence of spindle-like properties, such as Eg5-dependent microtubule flux (or minimally midzone localization of Eg5 or some other marker of antiparallel microtubule overlap).

Using our newly acquired larger data set of $n = 173$ z-stacks of bulk spindles with labeled kinetochores, we now provide data that correlates spindle polarity with both chromatin area (**Fig. 6d**) and number of chromosomes (**Fig. 6e**). We find that the number of spindle poles increases with both increasing chromatin area and number of chromosomes, which we now discuss in the main text (**Lines 303-311**). This analysis could not be done for the anchored chromosomes case since all the kinetochores are not resolved in the TIRF region, as previously mentioned. We also note that since we have better statistics, the majority (> 50%) of all bulk spindles imaged are now bipolar (**Fig. 6c**).

As requested, we developed a new immunofluorescence protocol (**Lines 282-302, 716-741**) and used it to stain our reconstituted spindles for both NuMA and Eg5. We observe sharp localization of NuMA to the spindle poles, as expected (**Fig. 6a, Fig. S5a**), allowing us to unambiguously count and identify poles. We also observe strong localization of Eg5 to the entire spindle with an enrichment at the poles (**Fig. 6b**), which is the expected localization for metaphase *Xenopus* egg extract spindles (Helmke and Heald, *J Cell Bio*, 206, 385-393, 2014, Fig. 4b). A negative control staining a random IgG showed no significant or distinct localization to the spindle (**Fig. S5b**). Thus, we conclude that our reconstituted system is indeed capable of forming bonafide bipolar spindles, with the proper localization of the dynein adaptor NuMA and the kinesin-5 Eg5.

Reviewer #2 (Remarks to the Author):

The authors have done a thorough job responding to my comments and the revised manuscript is greatly improved. I think that this work makes an important contribution to the field. After my readthrough of the revised manuscript, I have a few final comments aimed at increasing the clarity of the manuscript (mostly edits to the figures to improve the data presentation). With these changes, I am supportive of publication of this interesting work.

We thank the reviewer for their comments. We have made the requested changes to the revised manuscript as suggested below, which has resulted in clarifying each respective point.

Specific points:

- The authors did a good job altering the text to better emphasize/explain that de novo microtubule nucleation occurs, to clarify that not all microtubules in this system are templated off other microtubules. However, there was one remaining sentence that I still thought was confusingly worded (lines 116-117). Instead of stating that “These results demonstrate that microtubule nucleation in this system is microtubule-dependent”, it would be more precise to change it to “most microtubule nucleation in this system” or something similar (since microtubules can be nucleated de novo, not all nucleation is microtubule-dependent).

We have made this change (**Line 119**).

- Reviewer 1 requested that all figures include individual greyscale representations for each channel – I strongly agree with this suggestion. The authors did alter their figures in response to Reviewer 1's comment, but the single color images they included are not greyscale (they are red on black, green on black, etc). These colored panels are much harder to interpret than greyscale images (the contrast is less obvious so it is hard to see the signal). I strongly encourage the authors to change all single-color images to greyscale (in every figure), and only use colors in the merges.

We appreciate this suggestion and will make this change in the final proofing stage of the manuscript submission process.

- In figure 1B, 5A, and S1, the numbers in the bottom row (that denote unique microtubule plus ends) are very small and hard to see - make them bigger if you want to include them.

We have made this change to the main figures.

- The text is very small in some of the figures – I suggest going through and increasing the font size where possible. Some examples include the labels on the axes of graphs,

the text in Figure 4A, the labels in Figure 5A, the text in figure 6B-E, and the labels in Figure S1. The size of the entire Figure S3 could also be increased.

We have now made these changes.

- The graph in Figure S3 has four traces but only three conditions noted on the graph legend. I am assuming based on the colors that two of these are control traces, but the figure legend does not explain why two controls are shown. Please clarify this somewhere (and also clarify why the two control traces are different).

In Fig. S3, three data sets of average microtubule mass over time are plotted. The IgG positive control (black), the augmin immunodepletion (red), and the TPX2 immunodepletion (blue). These three data sets are marked in the legend. For each data set, multiple traces are plotted, where each trace is an independent extract experiment where multiple chromosomal branched networks are averaged. We have now better explained this figure in the Fig. S3 legend.

- I found the layout of Figure S6A to be confusing. The legend states that the bottom right panels show merged images, but many of the other images appear to be merges as well. I think that it would be helpful to alter the organization and labels on this figure, to make it more clear what is being shown in each image (and edit the figure legend to more clearly explain this as well).

We have now altered the presentation of Fig. S6A and changed the corresponding figure legend to enhance clarity.

- In some of the sections that were edited, new references were added, but these were not added into the numbered reference list (they are just listed as "Author, et.al." and not given a number or included in the reference list).

We have now appropriately referenced all citations.

Reviewer #3 (Remarks to the Author):

The authors' response addressed all of my points.

We thank the reviewer for their previous comments and appreciate the good news.